# A habit and working memory model as an alternative account of human reward-based learning

**Anne G. E. Collins** [1,2] ✉

Reinforcement learning (RL) algorithms have had tremendous success accounting for reward-based learning across species, including instrumental learning in contextual bandit tasks, and they capture variance in brain signals. However, reward-based learning in humans recruits multiple processes, including memory and choice perseveration; their contributions can easily be mistakenly attributed to RL computations. Here I investigate how much of reward-based learning behaviour is supported by RL computations in a context where other processes can be factored out. Reanalysis and computational modelling of 7 datasets ($n = 594$) in diverse samples show that in this instrumental context, reward-based learning is best explained by a combination of a fast working-memory-based process and a slower habit-like associative process, neither of which can be interpreted as a standard RL-like algorithm on its own. My results raise important questions for the interpretation of RL algorithms as capturing a meaningful process across brain and behaviour.

The reinforcement learning (RL) framework in computational cognitive neuroscience has been tremendously successful, largely because RL purportedly bridges behaviour and brain levels of analysis[1,2]. Model-free RL algorithms track the expected value of a state and update it in proportion to a reward prediction error[3]; this interpretable computation also accounts for important aspects of dopaminergic signalling and striatal activity[4,5]. Indeed, extensive research has supported the theory that cortico-striatal networks support RL-like computations for reward-based learning, and that disruption of this network causes predicted deficits in behaviour[6,7]. In parallel, similar model-free RL algorithms have been broadly and successfully used to explain and capture many aspects of reward-based learning behaviour across species, from simple classical conditioning[8] to more complex multi-armed contextual bandit tasks[9,10].

However, there is strong evidence that other cognitive processes, supported by separable brain networks, also contribute to reward-based learning[11,12]. Early research in rodents showed a double dissociation between so-called habits (thought to relate to the RL process) and more goal-directed processes, which are more sensitive to knowledge about the task environment and thus support more flexible behaviour[13–15]. Widely accepted dual-process theories of learning typically capture the slow/inflexible processes with model-free RL algorithms[16]. However, this apparent consensus hides broad ambiguity and disagreement about what the fast/flexible versus slow/inflexible processes are[17]. Indeed, recent literature has highlighted multiple processes that strongly contribute to learning. In more complex environments with navigation-like properties, this may entail the use of a map of the environment for forward planning[16]. Even in simple environments typically modelled with model-free RL, additional processes such as working memory (WM)[11], episodic memory[18,19] and choice perseveration strategies[20] have been found to play an important role. In particular, instrumental learning tasks such as contextual multi-armed bandits rely mostly on WM, with contributions of a slow RL-like process when load overcomes WM resources[21,22].

Because the RL family of models is highly flexible[3], RL models have nonetheless successfully captured behaviour that is probably more

[1]Department of Psychology, University of California, Berkeley, Berkeley, CA, USA. [2]Helen Wills Neuroscience Institute, University of California, Berkeley, CA, USA. ✉e-mail: annecollins@berkeley.edu

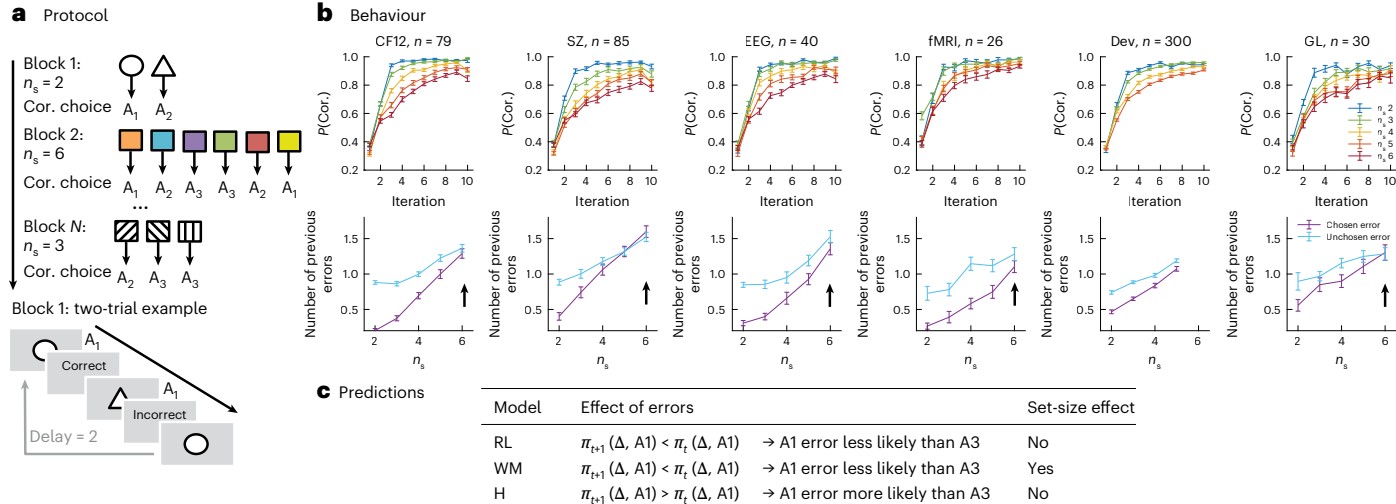

**Fig. 1 | Protocol, behaviour and predictions. a**, RLWM experimental paradigm. Participants performed multiple independent blocks of an RL task, using deterministic binary feedback to identify which of three actions was correct (Cor.) for each of $n_s$ stimuli. Varying $n_s$ targets WM load and allows me to isolate its contribution[21]. **b**, Behaviour (plotted as mean ± standard error) across six datasets on the RLWM task: CF12[21], SZ[24], EEG[31], fMRI[30], Dev[34] and GL (novel dataset). Top: learning curves showing the probability of a correct action choice as a function of stimulus iteration number, plotted per set size, illustrating a strong set-size effect that highlights WM contributions to behaviour. Bottom: error trial analysis showing the number of previous errors that are the same as the chosen error (purple) or the other possible error (unchosen; cyan) as a function of set size. The large gap in low set sizes indicates that participants avoid errors they made previously more often than from other errors; the absence of a gap in high set sizes indicates that participants are unable to learn to avoid their past errors (black arrows). **c**, Qualitative predictions for the RL, WM and H modules, based on the trial example in **a**. Only the WM module predicts a set-size effect[21]. Only the H module predicts that participants are more likely to repeat a previous error (for example, selecting action $A_1$ for the triangle) than to avoid it.

driven by other processes such as WM. Indeed, in most simple laboratory tasks, non-RL processes make very similar predictions to RL ones—for example, perseveration strategies might be mistaken for a learning rate asymmetry in RL[23], and WM contributions might be mistaken for high learning rates[21]. Non-RL processes become identifiable only in environments explicitly designed to attempt to disentangle them[18,21]. The contributions of non-RL processes to learning are thus often attributed to RL computations, and this misattribution of various processes to RL may lead to confusion in the literature, when findings relying on RL modelling are mistakenly attributed to RL brain processes[24,25].

Here I investigate how much of reward-based instrumental learning actually reflects a model-free RL process, as typically formulated in the literature. Because of the well-characterized and major contributions of WM in instrumental learning, I focus on a task context where WM's contribution can be adequately parsed out, the RLWM paradigm[21]. I parse out WM contributions to learning by its key characteristic: a strong limitation in resources or capacity[26]; note that this feature is not part of the typical characteristics of RL processes. I reason that a key characteristic of model-free RL is that it integrates reward outcomes over time to build a cached value estimate that drives policy directly, or indirectly through policy updates (for example, in actor–critic architectures[27]). More specifically, negative prediction error in model-free RL should make an agent less likely to repeat the corresponding choice. I thus focus here on how positive (correct, +1) and, more importantly, negative (incorrect, 0) outcomes affect later choices.

Behavioural analysis and computational modelling of seven datasets across two experimental paradigm versions (five previously published and one new for the deterministic version, RLWM; one previously published for the probabilistic version, RLWM-P) show that, when parsing out WM, we cannot detect evidence of RL in reward-based learning. Indeed, predictions including an RL process are falsified[28]. All behaviour can instead be explained by a mixture of a fast, flexible and capacity-limited process (WM) and a slower, broader process that tracks stimulus–action associations, irrespective of outcomes. Simulations show that neither process on its own can learn a reward-optimizing policy, and thus neither can be considered an RL process[3]; nonetheless, jointly as a mixture, the two non-RL processes do learn a good policy, supporting flexible human reward-based instrumental learning. These findings call for a reconsideration of how we interpret findings using the RL framework across levels of analysis.

## Results

The RLWM task was designed to disentangle the contributions of WM-dependent learning from those of slower, iterative RL processes to reward-based learning via manipulating information load. Across independent blocks, participants learned stable stimulus–action associations between a novel set of stimuli (the set size ($n_s$) ranged from two to six items within participants) and three actions. The correct action for each stimulus was deterministically signalled by correct (or +1) feedback, while the two incorrect actions were signalled with incorrect (or 0) (Fig. 1a). Participants' behaviour in low set sizes appeared close to optimal, but increasing set size led to increasingly incremental learning curves (Fig. 1b), a pattern replicated across multiple previous studies in diverse populations[21,22,24,29–37]. This pattern was uniquely captured by the RLWM model, a mixture model of two processes representing WM and RL. In this model, the RL process is a standard delta-rule learner, while the WM module has a learning rate of 1 to capture immediate perfect learning but also decay to capture WM's short timescale of maintenance; the mixture reflects WM resource limitations, such that behaviour is mostly driven by fast and forgetful WM when the load is within WM resources, but supplemented by RL with increasing load (Methods). This model included a bias weight parameterizing asymmetric updating of positive and negative feedback. This bias was shared between WM and RL and modulated learning rates for incorrect versus correct outcomes. Previous model fitting of the bias parameter (shared between WM and RL) revealed that incorrect outcomes had a weaker impact on subsequent choices than correct outcomes[34].

### Value and reward integration

To better identify the non-WM, set-size-independent, slower and incremental component of learning (putatively RL) in this task, I first sought

to understand how positive and negative outcomes were integrated to impact policy. Specifically, I reasoned that a process learning from reward prediction errors in an RL-like way should use negative feedback in error trials to make participants less likely to repeat mistakes, and more so the more they made the same mistakes (Methods and Fig. 1c). I thus computed, within error trials, whether the specific error participants made (out of two possible errors for a given stimulus) was indeed the one that had been made less frequently than the other error.

Across all six datasets in the RLWM task, the number of previous errors was overall lower for the chosen error than for the unchosen error (all $t > 4$, all $P < 10^{-4}$; Supplementary Table 1), showing that participants did use negative feedback overall in the task. As expected if participants' ability to use WM to guide choices decreased with set size, higher set sizes led to an increase in the number of previous errors for both chosen and unchosen. The difference between error type numbers, indicating participants' ability to avoid previously unrewarded choices, decreased with set size, as expected if higher set sizes reflected a higher portion of responsibility from a slower learning process (all $t > 2.28$, $P < 0.05$; Supplementary Table 2). However, I observed in all datasets that the difference decreased strongly (see the blue versus purple curves in Fig. 1b, arrows at $n_s = 6$), such that participants' policy appeared to become insensitive to negative outcomes selectively at set size $n_s = 6$ in four out of five datasets that included set size 6 (Supplementary Table 1). The effect even appeared to reverse in late learning in two datasets (Dev and SZ), such that errors committed late in learning in large set sizes had been repeated more often than the other error (all $t > 4.4$, $P < 10^{-4}$; Supplementary Table 3), showing error perseveration effects. I note that this pattern of error cannot be explained simply by increased noise with set size—indeed, a sufficient increase in noise to capture the observed error pattern would lead to much worse learning accuracy.

I compared participants' patterns of errors to the predictions from four variants of the RLWM model—one treating gains and losses equally in both WM and RL models, one with a shared bias[34] and the two best-fitting RLWM models with no or weak bias against errors in WM and full bias in RL, indicating complete neglect of negative outcomes in the RL module. All models captured the set-size effect of performance in the qualitative pattern of the learning curves (Fig. 2a), the main effect of the chosen versus unchosen error and the increase in the number of previous errors for both chosen and unchosen. The models also predicted that the difference between error type numbers (indicating participants' ability to avoid previously unrewarded choices) decreased with increasing set size. However, all models predicted that the difference should remain large even in large set sizes (see the blue versus purple curves in Fig. 2a; arrows at $n_s = 6$), contrary to what I observed empirically. In all six datasets, the magnitude of the difference decrease between the past numbers of chosen and unchosen errors could not be accounted for by any RLWM model, particularly late in learning (Fig. 2a, bottom, grey curves). Multiple other variants of models within the family of mixture models with RL and WM modules, relaxing some model assumptions or including other mechanisms, were tested but could not improve fits (Methods and Supplementary Fig. 2).

### The new WMH model explains behaviour

The behavioural and modelling results so far showed efficient integration of negative outcomes in low set sizes but not high set sizes, supporting the idea that WM uses negative outcomes to guide avoidance in policy, but the slower, less resource-limited process that supports instrumental learning under higher loads does not. However, even with an RL negative learning rate $\alpha_- = 0$, RLWM models could not capture the pattern, because WM contributes to the choices even in high set sizes where its contribution is diminished. Further variants of the RLWM family model, including those with policy-compression mechanisms, could not reproduce the qualitative pattern (Supplementary Fig. 6). I reasoned that the slow process should, to a degree, counteract WM's

ability to learn to avoid errors from negative outcomes. I thus explored a family of models where the slow module association weights ($Q$ values for RL) were updated with a subjective outcome $r_0$ for negative outcomes of $r = 0$. Surprisingly, the best-fitting model across six datasets (Fig. 3) was a model with fixed $r_0 = 1$, such that receiving incorrect feedback led to the same positive prediction error as correct feedback would. Negative learning rates still included a bias term shared across both modules. Note that this slow module cannot be interpreted as an RL module anymore, as the association weights track a relative frequency of stimulus–action choice, irrespective of outcomes, rather than an estimated value, and consequently the module cannot learn a good policy on its own. This module can be thought of as an associative 'Hebbian' or 'habit-like' module; thus, I label it H agent, with the mixture model WMH. While it is similar to a choice perseveration kernel[38], note that it is not purely motor but stimulus-dependent—indeed, all models also include a motor choice perseveration mechanism capturing participants' tendency to repeat actions across trials.

The WMH model fit quantitatively better than models with RL and WM (Fig. 3; see also Methods and Supplementary Fig. 3 for further models considered). It was also successful at producing the qualitative pattern of errors observed in real participants, such that errors at high set sizes appeared to fully neglect negative outcomes in a way that RLWM models could not (Fig. 2b, bottom; see Supplementary Fig. 6 for full validation of all models in Fig. 2a in all datasets). I further verified that this pattern of error changed dynamically over the course of learning in participants in a way that the model could capture (Fig. 2b, bottom).

### WMH also explains behaviour in a probabilistic reward learning task

While using the RLWM task was useful to adequately factor out WM contributions to reward-based learning, a downside is that the task does not necessitate the integration of reward in the same way probabilistic tasks do[6]. I thus sought to confirm whether my findings would hold in a probabilistic version of the task, RLWM-P; to that effect, I reanalysed a previously published dataset (see ref. 22, experiment 3). As previously reported, behaviour in this task was sensitive to set size ($F_{1,33} = 55.99$, $P < 0.001$; Fig. 4b), indicating that WM contributes to learning even in probabilistic environments thought to be more suited to eliciting RL-like computations. Similar to the deterministic task, I modelled behaviour with a mixture of two processes: a process capturing WM characteristics of set-size dependence and fast forgetting, and a process capturing the slower, non-forgetful and non-capacity-limited features (Methods). As in the previous datasets, the WM process model included RL-like equations; however, it is important to note that this process does not correspond to standard RL assumptions due to the strong capacity limitation. I compared mixture models where the slow process was either RL-like (that is, integrating negative outcomes differently from positive ones; RLWM) or association-like (that is, integrating negative outcomes similarly to positive outcomes). Supporting previous results, the best model was a WMH model including a fast, WM-like process that integrated negative outcomes as well as an outcome-insensitive, slower-learning component (Fig. 4a and Supplementary Fig. 10). This WMH model also fit better than the best single-process model and captured the qualitative pattern of learning curves (Fig. 4b, third panel from the left).

### RL-like policy with a simpler H algorithm

My results show that behaviour that is typically modelled with RL algorithms appears to instead be generated with non-RL processes, including a fast, forgetful and capacity-limited process that integrates outcome valence, and a slow and resource-unlimited H process that encodes association strengths between stimuli and actions, irrespective of outcome valences. This leaves two questions open: what is the computational function of this slow process, and why is it mistaken for value-based RL, for example in previous RLWM modelling[21,37]? Indeed,

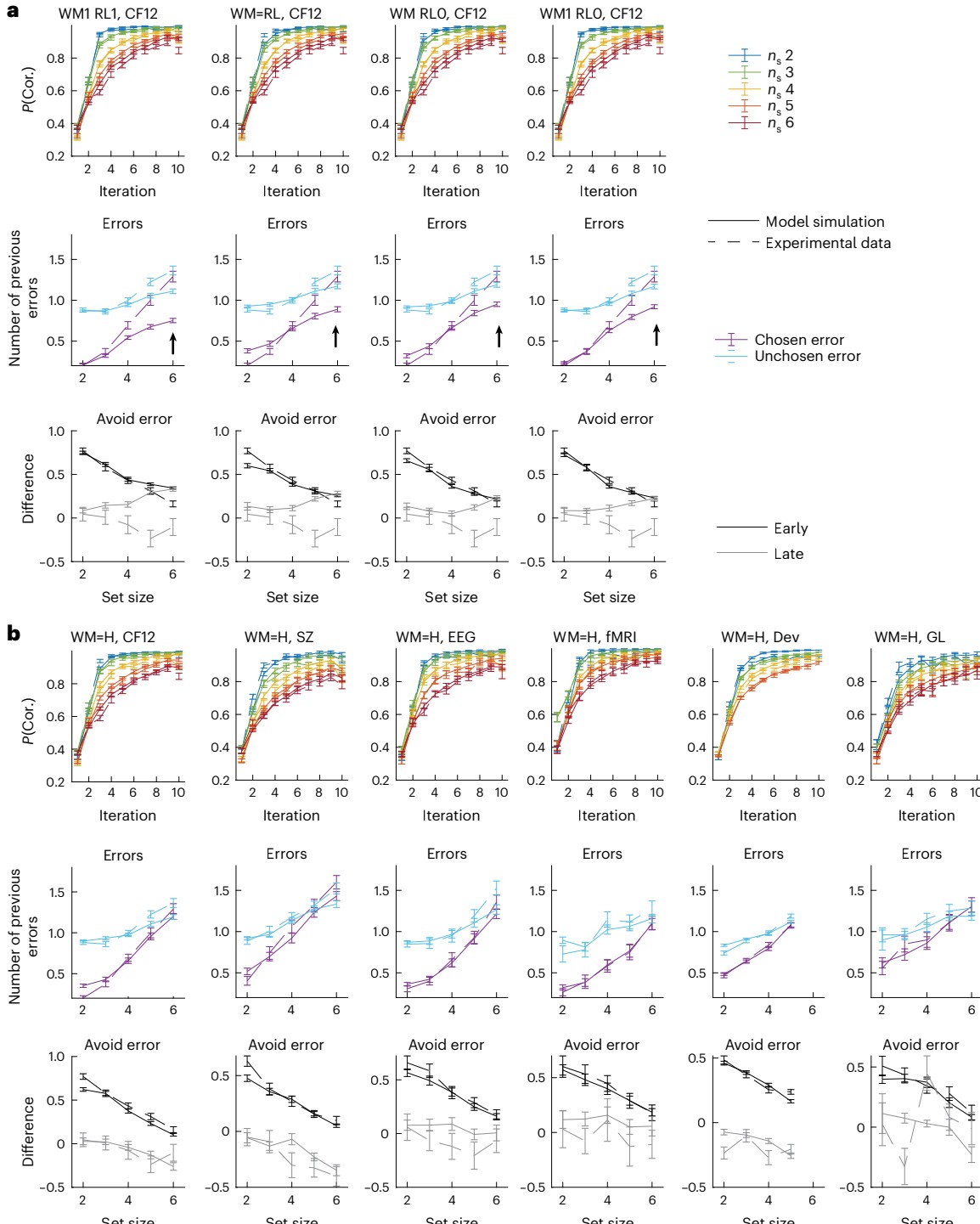

**Fig. 2 | Mixture models with WM and H capture errors better than mixture models with WM and RL. a**, Varying bias parameterization within the RL-WM family of models improves fit compared with previous models, by capturing the spread in learning curves better (top); however, the models cannot capture the pattern of errors (middle). The difference in past numbers of chosen and unchosen errors in error trials for early (iterations 1–5, black) versus late (iterations 6 and above) is not captured by any model (bottom). The models are illustrated on dataset CF12; see Supplementary Information for the other datasets. The dashed lines show the empirical data; the solid lines show the model simulations. **b**, The winning model WM=H captures patterns of behaviour better in all six datasets. The spread in learning curves across set sizes is better

captured (top). The new model captures the qualitative pattern of errors, such that in large set sizes, participants' errors are not dependent on their history or negative outcomes (middle). Neglect of negative feedback pattern differs in early (iterations 1–5) and late (iterations 6 and above) parts of learning; the WM=H model captures this dynamic (bottom). The models are indexed by their modules (WM, RL or H; Methods) and the bias term within their module (0 indicates $\alpha_- = 0$; 1 indicates $\alpha_- = \alpha_+$; no number indicates a free parameter; = indicates a shared free parameter). The data in all panels are plotted as mean ± standard error; the numbers of individual participants contributing to the plots for each dataset are indicated in Fig. 1.

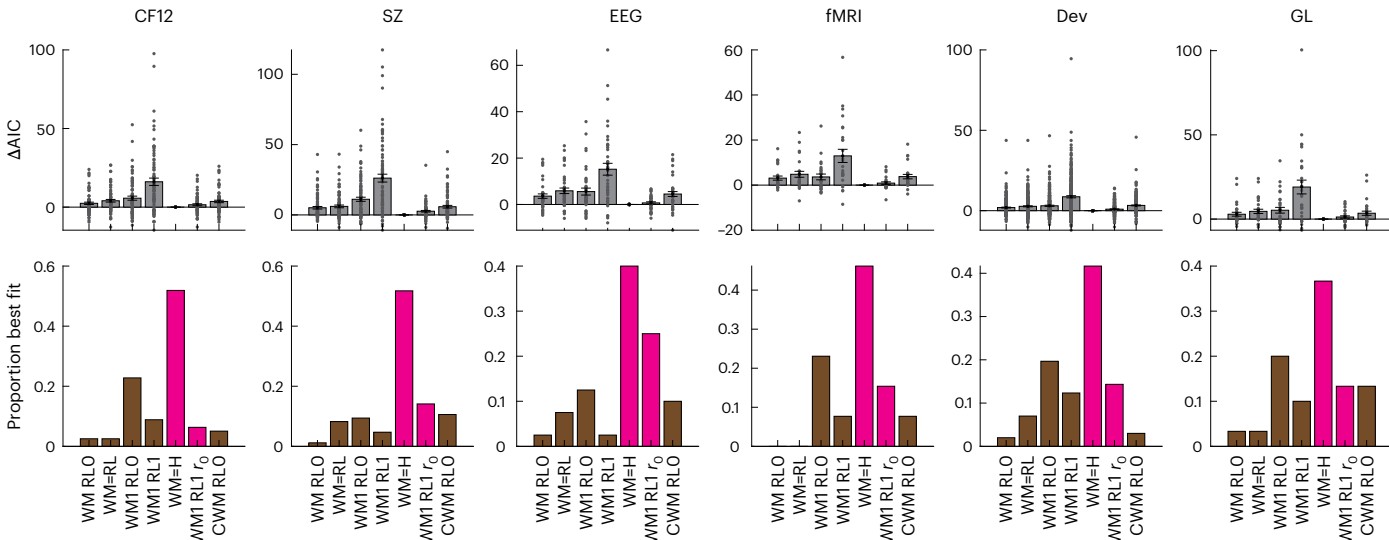

**Fig. 3 | Quantitative model comparison confirms a better fit for the WM=H than the WMRL family of models.** The top row shows individual (dots) and group mean AIC (± standard error), baselined to the group mean best model; the bottom row shows the proportion of participants best fit by each model. Both measures show that the WM=H model fits best in all datasets. $r_0$ indicates a free

parameter for the 0 outcome in RL; C indicates the use of policy compression. Results from models that can be interpreted as WMH are highlighted in pink and RLWM in brown. The numbers of individual participants contributing to the plots for each dataset are indicated in Fig. 1.

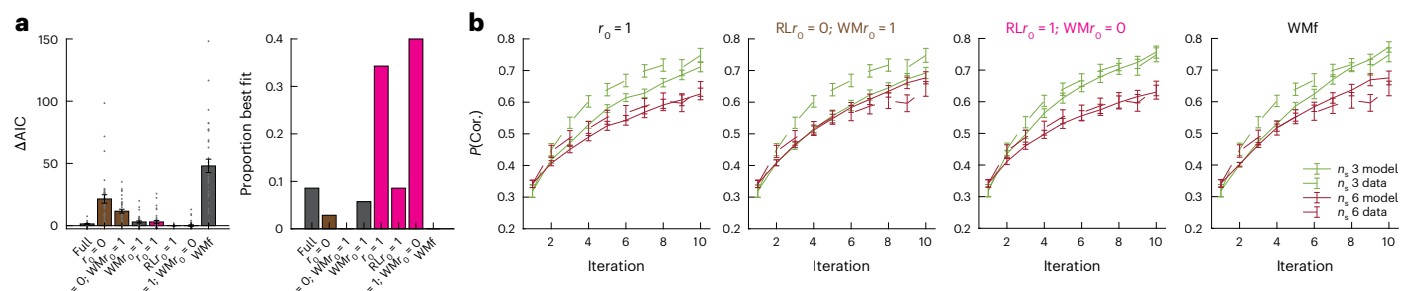

**Fig. 4 | Results replicate in a probabilistic learning task. a**, Model comparison showing the results from a family of models manipulating the subjective outcome value of outcome 0, $r_0$, for RL, WM or both—with $r_0$ a free parameter unless labelled to its fixed value. $r_0 = 0$ corresponds to standard RL or WM computations; $r_0 = 1$ corresponds to an H agent that handles both outcomes similarly. Highlighted in pink are agents that can be interpreted as WMH and in brown those that correspond to RL mixtures. The winning model $RLr_0 = 1$; $WMr_0 = 0$ assumes $RLr_0 = 1$ and $WMr_0 = 0$ and is thus a WMH agent, replicating the findings in the

deterministic version of the task. I further verified that the winning model was better than the best single-process model, WMf (Methods). The data are plotted as individual (dots) and group mean AIC (± standard error), baselined to the group mean best model; the right plot shows the proportion of participants best fit by each model. **b**, A set-size effect was also observed in a probabilistic version of the task; the winning model (third from the left) captures the learning curve pattern better than the competing models. The error bars indicate the standard error of the mean across $n = 34$ individual participants (dots in **a**).

on its own, the slow H process cannot learn a good policy but only tends to repeat previous actions, and thus seems functionally maladaptive. To investigate this question, I simulated both RLWM and WMH models in a standard probabilistic two-armed bandit task, varying the probability $p$ of a reward for the correct choice (Fig. 5, left, and Methods). RL policies track this value and thus convert to a graded policy where the agent is more likely to select the correct bandit at higher values of $p$ (green curve in Fig. 5, right). By contrast, an H agent on its own performs at chance, regardless of $p$ (blue curve in Fig. 5, right; mixture weight of the WM module ($\rho_{WM}$), 0). However, when the agents' choices invoke a mixture of policies, including a WM policy that tracks outcomes, the policy learned by the H agent does resemble a standard RL policy (dark blue curves). Indeed, even with low WM weights (for example, $\rho_{WM} = 0.5$), WM's contribution is enough to bootstrap choice selection of the good option, which leads the H agent to select this action more often and thus develop a good policy. This simulation shows that in the absence of specific task features decorrelating contributions of

rewards from contributions of errors to behaviour (such as the ability to consider multiple errors, something not feasible in most binary choice tasks), contributions of an H agent might be mistaken for an RL policy. Furthermore, in this mixture context, which probably corresponds to most human learning, I observe that the H agent does implement an adaptive policy with a simpler learning rule than the RL process.

## Discussion

I analysed six previously published datasets and one new dataset to investigate how different processes contribute to reward-based learning in humans. Such learning had previously been explained with model-free RL algorithms, which use a cached value estimate integrating past reward outcomes for given stimuli and actions to guide decisions. Behavioural analyses gave strong evidence across six datasets that the integration of outcomes to guide future decisions is dependent on load and becomes weak or absent at higher set sizes. My findings were present not only in healthy young adults but also in children ages 8–18,

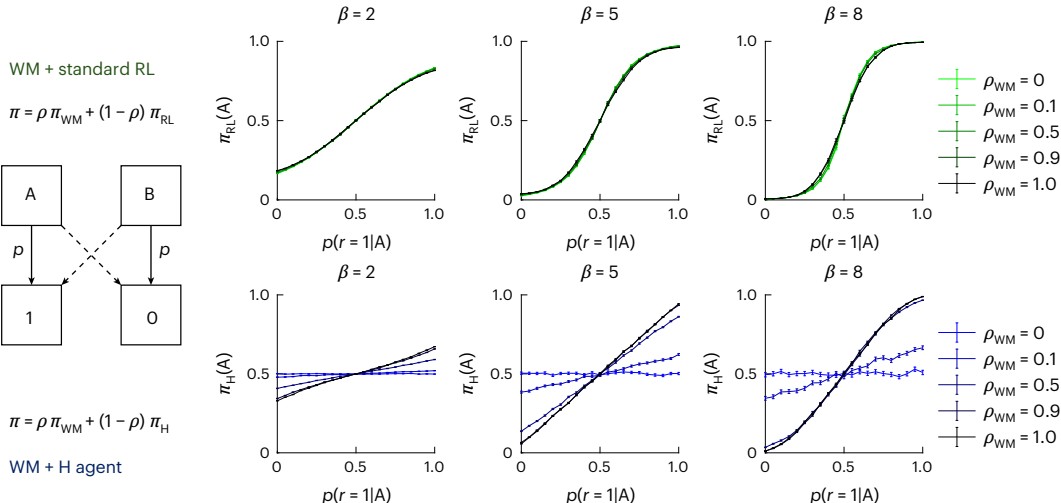

**Fig. 5 | H agents learn to mimic an RL policy when WM contributes to guiding choice.** Left: I simulated RLWM (top) or WMH (bottom) mixture agents on a simple probabilistic two-armed bandit task. Right: the policy learned by the H agent (bottom) resembles an RL policy (top) when there is enough WM contribution to choices, in a probabilistic two-armed bandit task. I varied parameters $\rho$ (indicating the contributions of the WM module) and $\beta$ (indicating the noise in the softmax policy). The error bars indicate the standard error of the mean across $n = 1,000$ simulations.

in healthy older adults matched to patients with schizophrenia and in patients, emphasizing the robustness of the findings across diverse populations. Computational modelling revealed that this pattern could be explained only by a mixture model, with two very distinct processes. The first, a WM-like process that learns fast but is limited in both the amount of information and the duration it can be held, appeared to successfully integrate reward outcomes into its policy. By contrast, a second, slower but less limited process appeared to fully neglect outcomes, updating in the same direction for wins and losses, and thus only tracked association strengths, in what could be likened to a Hebbian or habit-like process (H agent).

Although reward-based learning is, at first glance, well approximated by model-free RL algorithms, neither of these processes correspond to what is typically thought of as an RL cognitive process. The fast (WM) process integrates outcome values into a policy as an RL algorithm should, but it has properties not typically associated with RL, such as capacity limitations and rapid forgetting. By contrast, the slow, unlimited H process is more in line with what is typically thought of as RL along those dimensions, but it does not rely on reward-prediction errors—and indeed does not approximate values—as is typically expected from model-free RL algorithms in the context of cognitive neuroscience[3,39]. These processes also cannot, individually, be thought of as RL agents, in the typical sense of an algorithm that attempts to derive a policy that optimizes future reward expectations: on its own, the WM process can learn such a policy only under very minimal loads, while the H agent cannot learn such a policy at all.

I showed with simulations that the H agent, despite its learning rule that is on its own unsuited to learning from rewards, is nonetheless able to develop appropriate policies within a mixture model context. Indeed, using WM to bootstrap adaptive choice selection leads the agent to more frequently select actions avoiding bad outcomes, which further enables it to select good actions and reinforce them. This agent is mathematically equivalent to a stimulus-dependent choice perseveration kernel, which has been found to improve fit in other learning models[16,38,40], but is considered as an integral part of the learning mechanism rather than a low-level nuisance factor. In this way, my approach is reminiscent of the 'habits without value' model[41,42], which showed similar properties of developing good policies without value tracking. Here, my model extends the same theoretical approach to a stimulus-dependent learning context, and I experimentally validated the usefulness of this approach across seven datasets. The H agent

uses a simpler learning rule to learn a similar policy to an RL agent in a mixture context, which might be a more resource-rational way to lead to adaptive behaviour.

An important question concerns the generalizability of this finding to other learning tasks. Is it possible that the RLWM task, with deterministic feedback, incites participants to de-activate RL-like processes? While this is a possible explanation, I think it is unlikely. First, RL is not typically thought to be under explicit meta-control but rather to occur implicitly in the background[43,44]; thus, it is unclear why this would not be the case here. Second, computational modelling supports similar conclusions in the probabilistic version, RLWM-P, where integrating reward outcomes over multiple trials is useful, and H-like perseveration kernels have been found to improve fit in other probabilistic learning tasks[16,40]. Third, similar conclusions, using different methods, have very recently been drawn in different instrumental learning tasks in humans[45]. I limited my investigation here to the RLWM experimental framework because it offers a solid grounding for factoring out explicit WM processes and analysing what remains. However, an important future research direction is to find experimental and modelling approaches that will better allow us to parse out different processes, including WM, from learning behaviour, and to probe the generalizability of this finding to other instrumental tasks typically modelled with RL. A promising direction will be to systematically manipulate factors that decorrelate choice and reward history, allowing their separate contributions as well as their interactions to be investigated[46,47].

Another important question concerns the interpretation of the concept of RL across behaviour, algorithmic models and the brain mechanisms underlying the processes identified through modelling of behaviour. RL is a broadly used term, and ambiguity in its use across researchers can lead to confusion[25,48,49]. A reason for the success of model-free RL frameworks is their ability to map onto brain mechanisms in striato-cortical loops with dopaminergic signalling, including, for example, RL reward-prediction errors in striatal neural signal[50] (Supplementary Fig. 12). If learning from reward in humans appears RL-like to the first approximation but actually reflects two non-RL processes, how can we reconcile this with a wealth of RL-model-based neuroscience findings? I consider multiple possible explanations.

One possibility is that most of human reward-based learning tasks tap on WM processes that are at first approximation well described by RL (as here in the RLWM-P dataset), such that the striatal circuits support a more cognitive, explicit version of RL than typically assumed;

in parallel, the H agent might reflect Hebbian cortico-cortical associations[6]. Indeed, research in humans and non-humans has shown that model-free RL-like updates may occur over states and actions inferred through higher-level cognitive processes, leading to more flexible learning behaviour[12,17,35,48,51,52]. A second possibility is that model-free RL-like value learning in striatal-based networks does occur but does not strongly contribute to behaviour in many human experiments. Indeed, a three-process model (including WM, RL and H; Supplementary Fig. 3) fits worse than WMH in my datasets but can capture the qualitative pattern of errors. Thus, while in this dataset the RL component cannot account for error patterns, we cannot rule out a three-component model: it is possible that RL processes' contributions would be more evident under different experimental designs[29,53,54]. Nonetheless, recent findings show heterogeneity in striatal dopaminergic firing, with some patterns resembling H-like computations[55]. A third possibility is that the capacity-limited component of learning, which I attribute to WM processes here, is supported by the brain's RL network: that value-based learning does occur, but only within capacity limits and at a fast dynamical pace, fully supported by WM. Indeed, there is evidence for dopaminergic support of WM processes in both human and non-human animals[56–58]. This possibility would imply that RL processes are much more 'cognitive' than typically thought, and much more strongly dependent on capacity-limited WM than assumed. However, I think this explanation is unlikely: reward-based learning tasks that include a distant test phase in extinction where WM cannot contribute show results in line with standard accounts of RL, with choices depending on the incremental component in the learning phase[29–32].

Further research will necessitate careful task design, modelling and concurrent imaging to unconfound possible RL processes from other learning processes such as WM and H, and further our understanding of their neural correlates. Patient studies in the RLWM domain, including with lesion patients or patients with dopaminergic medications targeting striatum, should help shed light on these questions. Future research should also attempt to resolve possible concerns over model misspecification. Indeed, while WMH explains behaviour better than the competing models here, it is never possible to fully rule out other accounts. Thus, while I attempted to clearly delineate RL from non-RL predictions here, it remains possible that different implementations of RL not considered here might explain the pattern of errors[59,60].

My findings have important implications. First, they strengthen mounting evidence that RL modelling in reward-based instrumental learning tasks is useful but fraught[25,48,61]. While RL models capture much variance of learning behaviour, my findings hint that they do so often without actually capturing the dynamical cognitive processes that support behaviour. In addition to blurring our theoretical understanding, this may in practice lead to misinterpretations when RL models are used for model-based analysis of neural signals[30,53,62], or when fit RL parameters are used as mechanistically interpretable proxies for individual differences—for example, in developmental and ageing research[36,61,63] or computational psychiatry[24,29,37,64].

Second, my findings further highlight the fact that, beyond elegant, parsimonious single-process accounts of behaviour or broad dual-process ones, cognitive research has established a vast knowledge of multiple separable processes that support decision-making, including explicit memory processes such as WM. Even simple tasks designed to elicit a target process (such as bandit tasks for RL) recruit multiple other processes, but those processes may be unidentifiable in such tasks. Disentangling multiple processes requires considering more complex tasks to elicit differentiable behaviour. Future research in learning and behaviour should consider the parsimony/complexity trade-off carefully within the context of our knowledge of the complexity of human behaviour.

In conclusion, my findings in these datasets reveal that when learning from rewards, humans use effortful active maintenance of information to guide good choices in the short term, and rely on the iteration of choices over time to build a good policy, bootstrapped by limited memory. I found here no evidence of a standard value-based model-free RL contribution to learning and falsified the predictions of models that do include RL with standard assumptions. These findings call for care in interpreting any RL-based findings in instrumental learning domains, with important implications for behavioural, clinical, developmental and neurocognitive scientists.

## Methods

### Experimental design

All datasets were previously published[21,24,30,31,34], except dataset 6, GL. All studies were approved by an institutional review board (see publications), and the participants provided informed consent. All experiments relied on the RLWM protocol developed in ref. 21, with minor variations to the protocol across datasets. I first describe the shared components of the RLWM task and then describe specific details.

**Shared.** In all experiments, the participants' goal was to learn stimulus–action associations using truthful, binary feedback (correct/incorrect or +1/0). Actions corresponded to one of three adjacent key presses (or play console button presses). Each experiment included multiple independent blocks requiring learning for a novel set of easily identifiable stimuli.

Within each block, stimuli were presented for 10–15 iterations depending on the specific experiment, in an interleaved fashion. The number of stimuli (or set size $n_s$) was manipulated across blocks and varied between two and six; this key manipulation enabled me to affect load and thus identify WM contributions. The stimulus presentation order was pseudo-randomized to control for the delay between two successive iterations of the same stimuli, with a close-to-uniform distribution between 1 and $2n_s - 1$. This was important to identify the forgetting component of WM. The number of blocks ranged from 10 to 22 depending on the experiment.

Stimuli were presented for a short period (typically 1.5 s, depending on the specific experiment), during which the participant made a key press; this was followed by a short feedback interval (0.5–1 s) and then a short inter-trial interval (typically 0.5 s, but see the details of each published dataset). Stimuli within one block consisted of highly discriminable and familiar exemplars of a category (for example, a cat, a cow, an elephant and a tiger in the animal category).

The participants' instructions fully described the structure of the task, including the fact that feedback was truthful and correct stimulus–action associations did not change within a block. The participants were compensated for their participation either with cash or with course credits; see the publications for details.

**Published datasets 1–5.**

- Dataset CF12 (ref. 21) included $n = 79$ (44 female, ages 18–24, mean 24.3 ± 5.7 years) participants who performed the RLWM task in person. Set sizes ranged from two to six, for a total of 18 blocks.
- Dataset SZ[24] included $n = 85$ participants who performed the RLWM task in person, including patients with schizophrenia ($n = 49$) and matched controls ($n = 36$). Demographic information is available in Table 1 of Collins et al.[24]. To accommodate patients, the trial dynamics were slower; to keep the task within a shorter duration, the number of blocks was decreased to 13. See the published methods for the details.
- Dataset EEG[33] included $n = 40$ (28 females, ages 18–29) participants who performed the RLWM task in person while wearing an electroencephalography measurement cap. There were 22 blocks.
- Dataset fMRI[21] included $n = 26$ (11 females, ages 18–31 years) participants who performed the task in the scanner. To accommodate functional MRI timing constraints, the inter-trial interval durations were jittered, resulting in fewer blocks (18).

- Dataset DEV[34] included $n = 300$ participants (ages 8–30) who performed the task in person. To accommodate younger participants, the maximum set size was $n_s = 5$, and the number of blocks was reduced to 10. The participants used a game console with three buttons instead of a keyboard. Demographic details are available in Master et al.[34].

**Dataset 6 (GL).** The study protocol was approved by the University of California, Berkeley, Institutional Review Board. In this unpublished dataset, $n = 30$ (20 female, ages 18–25) participants performed a variant of the experiment where half of the blocks were 'gain' blocks and half were 'loss' blocks. In gain blocks, the participants tried to gain points, using feedback +1 versus 0. In loss blocks, the participants tried to avoid losing points, using feedback 0 versus −1 for the correct choice versus the two incorrect ones for each stimulus. There were 18 blocks, and 15 iterations per stimuli. I observed no difference in behaviour between the gain and loss blocks, and computational modelling did not uncover any differences either (that is, making any parameter from the winning model dependent on block condition did not improve fit). For the purpose of behavioural and modelling analyses in this paper, outcomes 0/−1 in the loss blocks were treated as correct/incorrect in the same way as outcomes 1/0 in the gain blocks.

**RLWM-P experiment.** The RLWM-P experiment was a variant of the RLWM experiment with probabilistic feedback. Previous analysis confirmed a set-size effect, showing WM involvement even when learning in a probabilistic context (experiment 3 in ref. 22). In this experiment, selecting the correct action led to positive feedback with probability $p = 0.92$ or $p = 0.77$ across blocks, while selecting the incorrect action led to negative feedback with the same probability. The participants ($n = 34$, 20 females, mean age 20.97 years) were informed of the probabilistic nature of the task. The participants experienced only two set sizes across 14 blocks (8 for $n_s = 3$ and 6 for $n_s = 6$) with 12 iterations per stimuli.

**Participants.** All procedures were approved by institutional review boards where data was collected (including the Committee for the Protection of Human Subjects at the University of California, Berkeley, for unpublished dataset 6, GL). The participants provided informed consent and were free to stop participation at any time of their choosing. Please refer to the corresponding publications for further participant and procedure details.

### Behavioural analysis
**Set-size effects on accuracy.** I visualized the data for each dataset using the same learning curve as in previously published analyses, where the average choice accuracy is plotted as a function of the specific stimulus iteration number, separately for each set size.

**Error analysis.** To investigate the effect of negative outcomes on behaviour, I designed an error trial analysis. I reasoned that if participants integrated negative feedback into their policy, they should be less likely to repeat a previous error. There were two possible errors for each stimulus (for example, if $A_2$ is correct for the triangle stimulus in Fig. 1a, then $A_1$ and $A_3$ are possible errors; the $A_3$ error should be more likely after $A_1$ is tried and results in incorrect feedback). Thus, if a participant performed an error $E_t$ for stimulus $S_t$, I counted how many times the participant had made the same error for stimulus $S_t$ up to trial $t − 1$ (chosen error) and how many times they had made the other possible error (unchosen error); this corresponds to the blue and purple curves in Figs. 1 and 2. To measure success at avoiding error, I also computed the average error avoidance success by subtracting the number of previous unchosen errors from the number of previous chosen errors (black and grey curves in Fig. 2).

### Computational modelling
**Model fitting.** I used MATLAB v.2020B with fmincon optimization to fit all computational models, with ten random starting points per participant and capacity (for discrete capacity models). I sought parameters that optimized the log-likelihood of the data under the model assumptions[65], fitting data from each participant independently. The parameter constraints for model fitting are as follows:

- Learning rates $\alpha$, bias parameters, decay $\phi$, mixture weights $\rho$, the noise parameter $\epsilon$ and the $r_0$ parameter were all constrained to their natural range of [0, 1].
- Motor perseveration parameters were constrained to [−1, 1], enabling both tendencies to repeat motor choices and to avoid previous choices.
- The capacity parameter was fit as a discrete parameter $K \in \{2, \dots, 5\}$ to avoid optimizer slowness with non-smooth likelihoods with continuous $K$ parameters; my previous experience comparing discrete versus continuous $K$ coding showed no difference[36,66] and that using discrete coding is a pragmatic choice rather than a theoretical commitment of the model. Furthermore, the upper and lower bounds on the capacity reflect both a strong theoretical prior about humans' ability to actively hold information in mind[26] and a pragmatic identifiability constraint: capacities higher than the maximum set size all lead to the same likelihood value. This constraint does not impact interpretation: should participants indeed have a higher capacity, then a single-process model should fit their behaviour better, which I can rule out through model comparison.

**Model comparison.** *Akaike information criterion.* For model comparison, I used Akaike information criterion (AIC)[67]. I have observed in the past that another widely used criterion, BIC, strongly overpenalizes complexity for models in the RLWM family, and I observed this again here via conduction model recovery analyses (see below). In figures, I report both the mean (and standard error) of AIC across the group (within each dataset) and the proportion of participants best fit by each model. Where comparable (datasets 1–6), I observed highly convergent best results (Supplementary Fig. 1).

*Model space exploration.* Because of the breadths of potential model space, I limited model space exploration to sequential families as described below. I performed model comparison within a model family, selected the best model out of each family and then performed model comparison again between winning models.

*Model validation.* To validate the winning model versus competing models, I simulated winning models with fit parameters, with 20 simulated agents per participant. Summary statistics of interest (for example, learning curves and error analysis) were averaged over agents within participants first, to average out stochasticity in simulations. I then plotted the resulting synthetic dataset behaviour across participants in the same way I plotted participants' behaviour (including mean and s.e.m. across synthetic participants).

**Checks.** *Model identifiability.* I performed model identifiability analyses within the key models of interest that represent theoretically interesting contrasts[65]. I ensured that model comparison via AIC was appropriate and that competing models were identifiable with confusion matrices (Supplementary Figs. 4 and 10).

*Parameter identifiability.* I performed parameter identifiability via generate and recover procedures for the model parameters; see Supplementary Figs. 5 and 11. The best-fit parameters are reported in Supplementary Figs. 7–9.

### Computational models—RLWM
**Mixture model.** Previous work[21,24,30–34,53] showed that behaviour in the RLWM task cannot be adequately captured with a single-process model. I used the RLWM modelling framework as a baseline, which assumes

that policy is the mixture of a WM policy, designed to capture fast but forgetful information integration, and a non-forgetful integrative process, typically RL:

$$\pi_{\text{mixture}}(a|s) = \rho_{\text{WM}}\pi_{\text{WM}}(a|s) + (1 - \rho_{\text{WM}})\pi_{\text{other}}(a|s)$$

where $a$ is the action, $s$ is the stimulus and 'other' is typically RL. The mixture weight $\rho_{\text{WM}}(n_s)$ is set-size dependent and serves to capture resource or capacity limitations of the WM process. In the context where set size is $\in \{2, \ldots, 6\}$, the mixture weight is set to $\rho_{\text{WM}} = \rho \min(1, K/n_s)$ where $K \in \{2, \ldots, 5\}$ is a capacity parameter, and $\rho \in [0, 1]$ regulates the overall balance of WM versus non-WM in the policy. If there are only two set sizes, the mixture weight is parameterized per set size ($\rho_{\text{WM}} = \rho_3, \rho_6$).

This full policy is typically mixed with a uniform random policy to capture random lapses in choices to produce the final full policy, with noise parameter $\epsilon \in [0, 1]$:

$$\pi(a|s) = (1 - \epsilon)\pi_{\text{mixture}} + \epsilon\frac{1}{n_A}$$

Note that other dual-process approaches (for example, ref. 16) perform the mixture at the value level rather than the policy level; this is because both processes are assumed to track comparable variables in those approaches (for example, estimated value). Here, instead, my two processes do not track directly comparable variables (RL value versus WM association weights), and as such, a mixture at the policy level is more appropriate.

**WM module.** The WM module tracks information in an association weight matrix initialized at the beginning of each block at $W_0 = 1/n_A$ reflecting the initial expectation that one out of $n_A = 3$ actions leads to reward 1 (versus 0). After observing stimuli, actions and rewards ($s_t, a_t, r_t$) at trial $t$, the update is

$$W_{t+1}(s_t, a_t) = W_t(s_t, a_t) + \alpha_{\text{WM}}(r_t)(r_t - W_t(s_t, a_t))$$

To capture one-shot encoding of information, I set $\alpha_{\text{WM}}(1) = 1$. To capture potential neglect of negative outcomes, I set $\alpha_{\text{WM}}(0) = \text{bias}_{\text{WM}}$ as a parameter, which is either free ($\text{bias}_{\text{WM}} \in [0, 1]$) or fixed depending on the model considered. To capture short-term maintenance in WM, WM weights are decayed at each trial towards initial values for all $(s, a)$ not observed at $t$:

$$\forall (s, a), W_{t+1}(s, a) = W_t(s, a) + \phi_{\text{WM}}(W_0 - W_t(s, a))$$

where $0 \leq \phi_{\text{WM}} \leq 1$ is a decay rate parameter.

The WM policy transforms the WM weights through a standard softmax:

$$\pi_{\text{WM}}(a|s) = \frac{\exp \beta W(s, a)}{\sum_i \exp \beta W(s, a_i)}$$

where the temperature parameter $\beta$ is typically fixed to a high value (here $\beta = 25$) for theoretical reasons (this ensures that the WM policy of a repeated trial is perfect) and identifiability reasons (this ensures that the RL learning rate is identifiable and the RL and WM modules are separable). In the absence of a free $\beta$ parameter, noise in the choice policy is instead parameterized as lapses in the overall policy via parameter $\epsilon$, which is highly recoverable (Supplementary Fig. 5).

**RL module.** The RL module is a standard delta-rule agent that tracks $Q$ values for each stimulus and action pair. $Q$ is initialized at $Q_0 = 1/n_A$, reflecting the initial expectation that one out of $n_A = 3$ actions leads to reward 1 (versus 0). The delta-rule update is:

$$Q_{t+1}(s_t, a_t) = Q_t(s_t, a_t) + \alpha_{\text{RL}}(r_t)(r_t - Q_t(s_t, a_t))$$

The positive learning rate parameter $\alpha_{\text{RL}}(1) \in [0, 1]$ is free, and the negative learning rate $\alpha_{\text{RL}}(0) = \text{bias}_{\text{RL}} \times \alpha_{\text{RL}}(1)$ is also parameterized by a bias parameter ($\text{bias}_{\text{RL}} \in [0, 1]$), which is free or fixed depending on the specific model.

The RL policy transforms the $Q$ values through a standard softmax:

$$\pi_{\text{RL}}(a|s) = \frac{\exp \beta Q(s, a)}{\sum_i \exp \beta Q(s, a_i)}$$

The temperature parameter $\beta$ is fixed and shared with the WM module (see above).

**RL-like module extension (H agent).** I extended the RL module to new versions of the algorithm to capture the observed error effects that standard RLWM models cannot capture.

Specifically, the H module tracks association weights in a way very similar to an RL module, and is initialized also at $H_0 = 1/n_A$. The update is:

$$H_{t+1}(s_t, a_t) = H_t(s_t, a_t) + \alpha_{\text{H}}(r_t)(\text{SR}(r_t) - H_t(s_t, a_t))$$

The only difference is the subjective outcome SR, which is fixed at $\text{SR}(1) = 1$ for correct outcomes and parameterized at $\text{SR}(0) = r_0$ for incorrect outcomes, with the parameter $r_0 \in [0, 1]$, free or fixed depending on the model. With $r_0 = 0$, the H agent reduces to an RL agent. With $r_0 = 1$, the H agent treats correct and incorrect outcomes exactly identically and increases the weights of the selected action no matter the outcome, thus only tracking a function of stimulus–action associations. The learning rate $\alpha_{\text{H}}$ is parameterized in the same way as $\alpha_{\text{RL}}$. The H policy transforms the $H$ values through a standard softmax:

$$\pi_{\text{H}}(a|s) = \frac{\exp \beta H(s, a)}{\sum_i \exp \beta H(s, a_i)}$$

The temperature parameter $\beta$ is fixed and shared with the WM module.

H agents replace RL agents in the standard RLWM mixture policy to form WMH mixtures:

$$\pi_{\text{mixture}}(a|s) = \rho_{\text{WM}}\pi_{\text{WM}}(a|s) + (1 - \rho_{\text{WM}})\pi_{\text{H}}(a|s)$$

**Choice kernels.** I explored including different choice kernels in the policy to investigate whether it improves model fit and to ensure that such choice kernels cannot account for the observed effects. I incorporated the choice kernels in both policies in the mixture.

*Sticky choice.* Sticky choice captures stimulus-independent choice perseveration—that is, the tendency to repeat the same key press in consecutive trials. Specifically, I implemented it within the softmax policy as:

$$\pi_{\text{WM}}(a|s) = \frac{\exp(\beta W(s, a) + \kappa I(a, a_{t-1}))}{\sum_i \exp(\beta W(s, a_i) + \kappa I(a_i, a_{t-1}))}$$

where $I(a_i, a_j) = 1$ if $i = j$ and 0 otherwise, and $\kappa \in [-1, 1]$ captures a tendency to repeat or switch away from the previous key press. I applied the same approach to Q and H agents, with shared parameters.

*Regularization.* Policy compression adds a choice kernel that favours default actions, such as actions that are valid across more stimuli than others[68]. Specifically, I implemented it within the softmax policy as

$$\pi_{\text{RL}}(a|s) = \frac{\exp(\beta Q(s, a) + \tau\bar{\pi}(a))}{\sum_i \exp(\beta Q(s, a_i) + \tau\bar{\pi}(a_i))}$$

where $\bar{\pi}(a) = \text{mean}_i(\pi(a|s_i))$. I applied the same approach to WM and H agents with shared parameters.

**Model space.** The model space resulting from the factorial combination of all considered mechanisms is too large to explore. I first considered mechanisms that may absorb variance of no current theoretical interest and asked whether adding them to the starting, best-so-far RLWM model (based on ref. 22) could improve fit. Specifically, I validated that sticky choice and $\epsilon$ noise in the policy systematically improved fit across datasets, but policy compression did not (and could not capture qualitative patterns of behaviour; Fig. 2).

I thus explored two families of models systematically:

- I first systematically explored the RLWM model (including free $\kappa$ and $\epsilon$ parameters) with the bias parameters $\text{bias}_{RL}$ and $\text{bias}_{WM}$ free, fixed to 0, fixed to 1 or shared, for a total of ten models (Supplementary Fig. 1 for model comparison). The best two models of this family (WM RL0 and WM1 RL0) both have fixed $\text{bias}_{RL} = 0$ (thus no update in RL after negative outcomes) and $\text{bias}_{WM}$ either free or fixed to 1 (thus limited learning bias in WM). In particular, they outperform the published baseline RL=WM model where a single bias parameter is shared[34].
- I then systematically explored the WMH model with $r_0$ free or fixed to 0 (same as RL) or 1, and free or fixed bias parameters. The winning model has fixed $r_0 = 1$ (a pure H agent with subjective outcome SR(0) = SR(1)) and shared free parameter $\text{bias}_{WM} = \text{bias}_{H}$.
- I additionally explored adding a policy compression mechanism to all models; the winning model from the corresponding family is labelled with 'C'. This did not improve fit and could not explain error patterns.
- I also verified that two specific, theory-driven assumptions of RLWM did not unfairly penalize the RLWM model family by removing these assumptions. Specifically, in the deterministic task, $\alpha_{WM}(1) = 1$ is set to the maximum fixed value to capture the theory-driven assumption that WM can store perfect information about a trial in a one-shot way. I verified that letting $\alpha_{WM}(1)$ be a free parameter did not improve fit or explain the qualitative pattern of behaviour (Supplementary Fig. 2). Second, I endowed WM but not RL with forgetting, capturing the knowledge that WM processes have short dynamic timescales, while RL processes are typically assumed to be more temporally robust. I verified that letting RL processes also have forgetting[69] did not improve fit or explain the qualitative pattern of behaviour (Supplementary Fig. 2).
- To ensure that the error pattern was not driven by preference for a specific action, which could lead participants to tend to repeat that action irrespective of feedback, I extended the RLWM family with a fixed biased action policy $\pi_{\text{bias}}(a_i|s) = \pi_i$, parameterized two free parameters. This biased action policy replaced the uniform random policy in the overall agent, reflecting the assumption that participants would select preferred actions when lapsing: $\pi_{RLWM} \leftarrow (1 - \epsilon)\pi_{RLWM} + \epsilon\pi_{\text{bias}}$. This model family did not improve fit or explain the qualitative pattern of behaviour (Supplementary Fig. 2).
- To check the robustness of my finding that the RLWM family cannot capture the pattern of results, I removed my assumption that participants use WM in proportion to items within capacity, and replaced it with an assumption that participants use only RL when the load is above capacity; specifically, I set the mixture weight for WM to be 0 when $n_s > K$. This model family did not improve fit or explain the qualitative pattern of behaviour (Supplementary Fig. 2).
- I replicated my previous finding that a capacity-limited WM module is necessary by fitting an RLfH model, which included a fixed mixture of an RL module with forgetting and bias (as above) and an H module. This model fit significantly worse, as expected from my previous findings that the size of the set-size effect on behaviour cannot be captured solely through decay mechanisms (Supplementary Fig. 3).

- Finally, I verified that a three-module mixture model was not necessary to capture behaviour. In this model, the policy was expressed as a tripe mixture with $\pi(a|s) = \rho_{WM}\pi_{WM} + (1 - \rho_{WM})(\rho_H\pi_H + (1 - \rho_H)\pi_{RL})$. I explored a version with shared learning rate bias parameters across modules, and a version with no bias for WM and H and a free parameter bias for the RL module. This model did not improve fit (Supplementary Fig. 3) but could capture the qualitative pattern of behaviour, as expected since it includes both WM and H.

The models included in the model comparison Fig. 3 are listed below.

All include at least six free parameters for WM capacity $K$, WM weight $\rho$, WM decay $\phi$, noise $\epsilon$, sticky choice $\kappa$, and H or RL learning rate $\alpha_H$ or $\alpha_{RL}$:

(1) WM RL0: RLWM model with free $\text{bias}_{WM}$ and fixed $\text{bias}_{RL} = 0$. Total seven free parameters.
(2) WM=RL: RLWM model with free $\text{bias}_{WM} = \text{bias}_{RL}$. Total seven free parameters. This model corresponds to Master et al.[34] with an additional sticky choice mechanism, which improved fit.
(3) WM1 RL0: RLWM model with fixed $\text{bias}_{WM} = 1$ and $\text{bias}_{RL} = 0$. Total six free parameters.
(4) WM1 RL1: RLWM model with fixed $\text{bias}_{WM} = 1$ and $\text{bias}_{RL} = 1$. Total six free parameters. This model is the 'no bias' model.
(5) WM=H: overall winning WMH model with free $\text{bias}_{WM} = \text{bias}_{H}$. Total seven free parameters.
(6) WM1 RL1$r_0$: RLWM model with fixed $\text{bias}_{WM} = \text{bias}_{RL} = 1$ and free RL SR(0) = $r_0$. Total seven free parameters. This model captures qualitative behaviour similarly to the WM=H model, because the $r_0$ parameter is fit to a high value for most subjects. As such, the WM1 RL1$r_0$ model identifies subjects as WMH agents rather than RLWM agents.
(7) CWMRL0: the best model in the policy compression RLWM family, with seven parameters including free $\text{bias}_{WM}$ and $\tau$ parameters.

## Computational models–RLWM-P

The computational model for the RLWM-P model was also a mixture model, with a slightly different WM module and an identical RL/H module. In the deterministic experiment, the WM module approximates encoding of the trial information in WM by maintaining relative state–action association weights. In a probabilistic context, by contrast, it is possible that participants hold in mind a hypothesis about the best action rather than specifically the last trial information. I sought to incorporate this into an extended version of the WM module.

To approximate WM and contrast it to either an RL agent or an H agent, I included the following assumptions:

- I constrained $\rho_{WM}(n_s = 6) < \rho_{WM}(n_s = 3)$ as a theoretical interpretability constraint ensuring that the WM-labelled module is more expressed under a lower load.
- I included forgetting only in the WM module to associate variance captured with rapid forgetting to the WM-labelled process. This is not a theoretical commitment to RL/H agents not potentially also experiencing decay, but rather that any decay should be stronger in WM, and thus a pragmatic choice to enable identification of the modules.

With these constraints, I used the same formulation as above for WM weights, but I let $\alpha_{WM}$ be a free parameter, such that the WM module might remember only the last trial for a given stimulus–action–reward (if $\alpha_{WM} = 1$) but might integrate over a few trials otherwise, capturing hypothesis maintenance. In this sense, the WM module is approximated by an RL-like computation with decay and is forced to contribute more to $n_s = 3$ than $n_s = 6$.

The full model includes 11 parameters: one per module each of positive and negative learning rates $\alpha(r)$ (four total); two mixture weight parameters $\rho$, one decay parameter $\phi$, one noise parameter $\epsilon$, one perseveration parameter $\kappa$ and one SR(0) = $r_0$ parameter each (similar to the H module above).

To explore the model space, I first fit the full model and then fixed the $r_0$ parameter to 0 (standard) or 1 (H agent) in the WM or the RL module or both. The winning model had fixed $r_0$(RL) = 1 (pure H agent) and $r_0$(WM) = 0 (standard WM agent). I next verified that fixing any other parameter (including $\rho, \kappa, \phi, \epsilon$ or biases) to fixed values did not improve fit over the winning model. Last, I verified that the winning model fit better than a single-module model that included all mechanisms and differential noise per set size (WMf, Fig. 4). I performed model recovery and parameter recovery checks as previously described for RLWM (Supplementary Figs. 4, 5, 10 and 11).

## Simulations

**Environment.** To investigate the computational role of an H-like agent, I ran simulations of two mixture agents representing RLWM (a mixture of WM and standard RL) and WMH (a mixture of WM and no-outcome associative H agent) on a simple probabilistic two-arm bandit task. Agents chose between two options (A and B) for $T$ trials and received reward $r = 0/1$ with $P(r = 1|A) = p$ and $P(r = 1|B) = 1 - p$. I varied $p \in [0:0.05:1]$ and $T \in [20, 50]$. The results were similar for the two learning durations, so I only plotted $T = 50$. I investigated three values of the exploration softmax parameter $\beta \in \{2, 5, 8\}$.

**Model.** The agents made choices on the basis of the mixture model policy $\pi = \rho_{WM} \pi_{WM} + (1 - \rho_{WM} \pi_{H/RL})$. However, I was interested in the policy learned by the non-WM model in the presence of WM to guide choices, and thus plotted $\pi_{RL}$ and $\pi_H$ rather than $\pi$.

I approximated a WM process with a simplistic one-back memory process, such that after each choice $C_t \in \{A, B\}$ and outcome $r_t$, we updated a WM associations buffer with $W_{t+1}(C_t) = r_t$. This captures the last reward obtained for each choice and crudely captures a no-integration, resource-limited, short-term memory process. WM policy was derived through a softmax transform: $\pi_{WM}(C) \propto \exp(\beta W(C))$.

The standard RL agent tracked the value $Q(C)$ of each choice by updating with a standard delta rule: $Q_{t+1}(C) = Q_t(C) + \alpha(r_t - Q_t(C))$. The learning rate parameter was fixed to $\alpha = 0.1$. RL policy was derived through a softmax transform: $\pi_{RL}(C) \propto \exp(\beta Q(C))$.

The associative H agent tracked the association strength $H(C)$ of each choice by updating with an outcome neglect learning rule: $H_{t+1}(C_t) = H_t(C_t) + \alpha(1 - H_t(C_t))$. The learning rate parameter was fixed to $\alpha = 0.1$; the results were similar with other $\alpha$ values. H policy was derived through a softmax transform: $\pi_H(C) \propto \exp(\beta H(C))$.

## Data availability

All data are available via GitHub at https://github.com/AnneCollins/WMH.

## Code availability

All code is available via GitHub at https://github.com/AnneCollins/WMH.

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

## Acknowledgements

I thank members of the Collins lab for their feedback and support during the development of this work, as well as the authors of the previously published datasets reanalysed here. I was partially supported by NSF grant no. 202844 and NIH grant no. R21MH136528.

## Competing interests

The author declares no competing interests.

## Additional information

**Correspondence and requests for materials** should be addressed to Anne G. E. Collins.

