## [Peer Review File · Nature Human Behaviour]

A habit and working memory model as an alternative account of human reward-based learning

Corresponding Author: Dr Anne Collins

Version 0:

Decision Letter:

10th December 2024

Dear Dr Collins,

Thank you once again for your manuscript, entitled "RL or not RL? Parsing the processes that support human reward-based learning.", and for your patience during the peer review process.

Your Article has now been evaluated by 3 referees. You will see from their comments copied below that, although they find your work of considerable potential interest, they have raised quite substantial concerns. In light of these comments, we cannot accept the manuscript for publication, but would be interested in considering a revised version if you are willing and able to fully address reviewer and editorial concerns.

We hope you will find the referees' comments useful as you decide how to proceed. If you wish to submit a substantially revised manuscript, please bear in mind that we will be reluctant to approach the referees again in the absence of major revisions.

In particular, we believe that it will be necessary to carry out analyses of additional datasets to address Reviewer #3's point about generalizability.

If you wish to submit a suitably revised manuscript, we would hope to receive it within 4 months. I would be grateful if you could contact us as soon as possible if you foresee difficulties with meeting this target resubmission date.

- Include a "Response to the editors and reviewers" document detailing, point-by-point, how you addressed each editor and referee comment. If no action was taken to address a point, you must provide a compelling argument. When formatting this document, please respond to each reviewer comment individually, including the full text of the reviewer comment verbatim followed by your response to the individual point. This response will be used by the editors to evaluate your revision and sent back to the reviewers along with the revised manuscript.
- Highlight all changes made to your manuscript or provide us with a version that tracks changes.

Link Redacted

Thank you for the opportunity to review your work. Please do not hesitate to contact me if you have any questions or would like to discuss the required revisions further.

Sincerely,

██████████

████████████████████

████████████████████

Nature Human Behaviour

REVIEWER COMMENTS:

Reviewer #1 (Remarks to the Author):

This study represents the culmination of the author's extensive body of research on the role of working memory in reinforcement learning tasks. It provides important evidence that challenges the common assumptions about RL models in reward learning tasks, which have been widely accepted for the past two decades. The methods used in the study are solid and careful, and the results are meaningful. I believe this paper has the potential to make a big impact and is certainly worthy of publication in Nature Human Behaviour.

On the other hand, the current analysis relies on comparisons within a somewhat restricted model space and is dependent on the assumptions of the models. In order to provide more robust support for the author's claims, I would like to suggest several points for further investigation and discussion. In addition, I believe that a more careful discussion of how this study fits into the context of previous research would strengthen its contribution. I outline my suggestions below.

Major point #1

The author proposes a WM model where the learning rate for rewarded outcomes is fixed at 1, and the decay rate for unchosen values is controlled by a parameter, ϕ_{WM} . In contrast, the RL module treats the learning rate as a free parameter and assumes that unchosen values remain unchanged.

Several previous studies have proposed RL models that incorporate forgetting or decaying mechanisms, such as the forgetting-Q learning models, where unchosen values decay at the same rate as the learning rate, or where the decay rate is independent of the learning rate. These models have consistently been shown to fit behavioral data better than models without forgetting or decay (e.g., Ito & Doya, 2009; Gershman, Zhou, & Kommer, 2017; URLs listed below).

<https://pubmed.ncbi.nlm.nih.gov/19657038/>

<https://direct.mit.edu/jocn/article/29/12/2103/28731/Imaginative-Reinforcement-Learning-Computational>

How well would the author's data fit if an RL module incorporating such forgetting mechanisms is used? Would the WM module with a fixed learning rate of 1 and a choice kernel still outperform such models? Furthermore, if the author's claims are correct, does this imply that the results reported previously with forgetting-Q learning could be reinterpreted as reflecting WM processes? Could the improved fit of models with forgetting mechanisms be attributed to WM?

In order to clarify the position of this study within the broader context of RL research, I suggest that these points be explored and discussed.

Major point #2

The author's claims basically rely on the goodness-of-fit of the proposed models. While not strictly necessary, supporting these claims with model-neutral methods would strengthen their validity. According to the author, human choices in the task can be explained by two independent effects: a reward-dependent working memory (WM) component and a reward-independent perseveration component based on past choices. It would be ideal to demonstrate the independence of these components and the validity of each in a manner that does not depend on the specific model assumptions.

A simpler alternative might be to use a regression model with predictors representing the choice history at each lag and the reward history for each option (cf. Lau & Glimcher, 2005). This could show that rewards primarily influence recent trials while past choices have effects over longer lags. Additionally, confirming the absence of interactions between reward (given choice) and choice history would further validate the claim (cf. Katahira & Kimura, 2024).

<https://pubmed.ncbi.nlm.nih.gov/16596980/>

<https://link.springer.com/article/10.1007/s42113-022-00145-2>

In addition, if the coefficients for each lag of the choice history can be shown to decay exponentially, it would support the validity of the choice kernel assumption.

However, given the limited number of trials per stimulus in the current dataset, such analyses may lack sufficient data to produce stable estimates. At the very least, discussing these possibilities as directions for future research would be valuable for readers.

Major point #3:

The estimated value of the WM decay rate, ϕ_{WM} , is not explicitly reported, and the author should provide this information. Related to Major points #1 and #2, if ϕ_{WM} is close to 1, it could be interpreted as a special case of the Forgetting-Q learning model where the learning rate equals the forgetting rate. In such a case, the effects of past rewards at each lag might manifest only as independent main effects, with no interactions (e.g., no interaction between rewards from one trial back and two trials back).

If ϕ_{WM} takes a value different from α_{WM} , interactions between past histories would be expected to emerge. This distinction is important, as it directly relates to whether the effects of past rewards are additive or interactive (cf. Katahira, 2015).

<https://www.sciencedirect.com/science/article/pii/S0022249615000218>

Minor points

p.2

“for example, perseveration strategies might be mistaken for a learning rate asymmetry in RL (Sugawara & Katahira, 2021),”
- The original study making this claim is Katahira (2018), and it should be cited in this context.

<https://www.sciencedirect.com/science/article/pii/S0022249617302407>

p.18

In the Model fitting section, it is stated that `fmincon` was used to optimize the log-likelihood. However, it is not clear whether independent maximum likelihood estimation was conducted for each participant (without any prior). Since protected exceedance probability is calculated, this seems likely, but this assumption might not be obvious to all readers without prior knowledge.

In addition, if independent maximum likelihood estimation was performed, how were the log-marginal likelihoods (log-model evidence), required for computing the protected exceedance probability, calculated? It is possible that BIC was used as an approximation, but this should be explicitly stated for clarity (while some researchers use AIC, I think that AIC was not derived as an approximation of marginal likelihood and is therefore not suitable for this purpose).

p.20

The “s” representing stimulus in the equations is in mixed case.

p.25-

The number of significant digits in the tables should be standardized to an appropriate level to improve clarity and consistency. Additionally, the representation of p-values should adhere to standard conventions, such as reporting exact values (e.g., 0.104) or thresholds (e.g., < 0.001).

Reviewer #2 (Remarks to the Author):

Summary

This paper builds on a substantial body of prior work by Collins, demonstrating the role of working memory in reward-based learning. Previous studies suggested that when individuals solve reinforcement learning (RL) tasks (e.g., learning the correct action for a given stimulus through trial and error), their behavior reflects multiple processes rather than just simple RL algorithms (e.g., updating a stored action value for each stimulus according to a prediction error). In addition to the RL component, which learns slowly, a working memory component is also involved. This component is modeled as an RL agent with a high learning rate, subject to forgetting over time, and its influence is limited by set size. This new paper re-analyzes six datasets previously thought to support this model and re-investigates the contributions of the basic RL component, showing that it is better described as a perseverative associative process (reinforced in the direction of the previous action, regardless of whether that action was rewarded). The paper concludes that there is no simple or typical “RL” component in human reward learning, contrary to expectations based on dopamine and value updating theories.

Overall contribution

I'm somewhat undecided on the overall importance and information gain of the findings. In comparison to the prior body of work by the author, this new article argues that there is no incremental reinforcement learning component in human reward learning. Rather, there is just a working-memory-related RL component and also a perseverative component. This is taken to argue against simple reductionist accounts of RL which perhaps would assume that because dopamine represents reward prediction errors and neuroscience theories cast value learning as gradual synaptic updating, then behavior would also reflect this. I think this argument is quite important.

Having said that, I have some hesitations about the strongest arguments that this paper puts forward, and also what the importance is of these particular findings in advancing this argument. First, I don't think the findings show that RL models of human behavior are incorrect, or that humans don't do RL in solving these types of tasks. Rather, I think they simply show that human RL involves working memory. Second, the existence of a perseverative or “H” component already is a standard feature of models of human bandit learning. For example, this type of model is used in standard bandit tasks (e.g. Rutledge et al., 2009, *Journal of Neuroscience*) and also in the two-step task (Daw et al., 2011, *Neuron* - note here that the

perseverative component is not just for successive motor actions). Interestingly, the new model in this paper winds up looking more like the previous model in these papers - having a component that does RL with a somewhat high learning rate and also a perseverative component. In relation to these models, the new model in this paper adds that the RL process has WM characteristics.

I do think the demonstration that the only RL behavior that is detectable in these tasks is influenced by (and may be entirely constituted by) working memory processes is valuable. And I agree with the argument that this argues against a reductionist view that human RL behavior reduces to incremental synaptic-like updating by dopamine. To some extent, though, I think this is already appreciated by the field. For example, Wang et al., 2018, Nature neuroscience (prefrontal cortex as a meta-learning system), demonstrate computational models for how RL computations can take place by activity rather than weight-based changes (notably here, learning the RL strategy itself involves dopamine). More generally, in the field, with the rise of transformers and language models there is now an increasing appreciation that strategies like RL can have a multiplicity of implementations (in weight vs activation based like WM). Nevertheless, the current article is a nice demonstration in this direction. I wonder though whether the position it argues against, and the relative portrayed dominance of this position, is somewhat overstated.

I'll now move to specific critiques of the analysis.

Analysis critiques

1. It is worth noting that the perseverative component of RL models can sometimes result from mis-specification of the RL model. For instance, Findling/Skvortsova et al. (2019, Nature Neuroscience) demonstrate that noisy RL updates to values can produce behavior that appears perseverative (e.g., behavioral correlations across successive decisions). Can this be ruled out as a reason for perseverative behavior here?
2. Relatedly, can a three-component model that includes RL in addition to WM and H components be ruled out?
3. Could there be general action biases for a stimulus (e.g. action 1 arbitrarily preferred for stimulus 1?)?. Could these mask as a perseverative effect?
4. Given that inferences about the slow process rely on factoring out the fast process, it is necessary to interrogate whether the WM process is correctly modeled. I have the following questions and suggestions about the WM part of the model. First, if participants are aware that there is only one correct action per stimulus, then I would think a WM component would do counterfactual updating (e.g., learning that action 1 is correct would imply action 2 is incorrect). Has this model been tried? Are results around the slow model robust to this?
5. It appears that capacity parameter K is constrained to be an integer. This seems unjustified. What happens if capacity parameters are allowed to vary continuously? More generally, I think K should also be unbounded. Based on the current setting, I think K being bounded at 5 prevents the WM system from ever fully turning off.
6. Related to the above, it is somewhat hard to understand why an RL agent with no learning from negative feedback cannot capture the basic error analysis effect. The paper reports that this is because the WM model remains active even at very high stimuli. This raises the question of why the WM component cannot turn off at a high number of stimuli. Could this be due to the parameter constraints mentioned above?
7. In the mixture model, averaging occurs at the level of the policy. In other similar mixture models (e.g., model-based / model-free by Daw et al., 2011), it is the value predictions of the two components that are averaged. What is the justification for averaging the policies rather than the values? Is this supported by model comparison?

Minor

8. Is the printed equation for the mixture weight a mistake ($p_{wm} = p \cdot \min(1, \frac{ns}{K})$)? I would think capacity should be divided by the set size (divided evenly among the items) rather than the other way around.
9. Please list the best-fitting parameters of the key models. It would be good to verify the fitted learning rates of each system.
10. For each model, please clearly list all parameter constraints in the fitting process and provide some consideration on how it is considered that such constraints do not affect interpretations.

Reviewer #3 (Remarks to the Author):

In this paper, Collins presents model-based and empirical evidence against a widely held assumption in instrumental tasks that model-free RL (defined as updating action values using a delta rule in response to RPEs) contributes to choice behavior. The author employs an error-based analysis across seven datasets, all from a paradigm designed to dissociate the contribution of working memory and model-free RL. The author also develops a novel alternative model to explain the error patterns. The key finding is that it is sufficient to include an outcome-independent habit mechanism, in addition to

outcome-sensitive WM, to explain the error patterns. I broadly agree with the author's point that even "simple" learning tasks mask the presence of other cognitive processes that we cannot measure, but which nevertheless contribute to choice behavior. The novel demonstration that when those are factored out, WM + habit is enough to explain behavior in this task is compelling, and will be of wide interest to the community studying related tasks. That said, I am not convinced that the findings generalize as broadly as the paper claims. Below are some comments and questions:

1. The term "RL" is overloaded in the paper/title. It is introduced, as is common, as a framework that can explain a variety of learning processes. But in this paper (and indeed maybe the field at large), RL is equated with a much narrower definition of model-free RL (e.g. on page 3) that updates values using a delta rule. Not being consistent & precise in terms of terminology might lead readers (especially those new to the field) to get the impression that RL as an explanatory framework is narrower than it actually is. In reality, there is broad consensus that, as the author clearly shows in the RLWM task, model-free RL on its own is insufficient to explain behavior.
2. The second suggestion is to consider what happens when we recast WM and other functions that work in service of model-based learning (e.g. attention), as internal actions (Dayan, 2012; Callaway and others in recent years). In this sense, actions are clearly not outcome independent, as success depends on using the correct model (i.e. loading efficient representations in WM, directing attention to relevant features, hypothesis sampling etc). Such actions might need to be learned (see work on attention learning from Kruschke, Niv and others). Can we confidently say, based on these data, that "mental actions" are not learned in a model-free way (Bramlage & Cortese)? Or that, once the correct model is considered, learning does not proceed in a model-free manner (Lee ... Witten, Daw)?
3. This representation learning reframe suggests that an alternative explanation for outcome insensitivity at large set sizes might be "meta-learning", e.g. people verbalizing and updating a concise model/construal/representation of the task (see work from Ho, Radulescu & others). This type of "efficient" model learning is exactly the kind that might be diverging in SCZ and development.
4. With regards to generalizing to probabilistic versions of task, maybe there is a bit of an overgeneralization here? The probabilistic version of RLWMP has fixed rewards probabilities, which means participants can still solve it using a policy that might take the form "when this symbol appears I should click this". OTOH Bornstein shows effects of episodic memory on choice in a drifting 4-arm bandit, which seems more convincing. In any case, in real-world learning scenarios, outcomes or optimal state-action mappings are rarely deterministic or fixed; instead, they often involve a mix of uncertainty and variability that challenges purely habit-based or one-shot responses and requires adaptive learning strategies. To strengthen the findings, it would be valuable to test a broader range of task conditions with varying set sizes and outcome stochasticity to understand the interaction between set size and state uncertainty.
5. Within the context of the data in this paper, one idea would be to compare models across each set size separately. For instance, smaller set sizes (1 to ~3) might involve more one-shot learning, which might make the WMH model fit better due to task demands, rather than an absence of RL-like processes. This would allow a more precise view of strategy use across different task conditions.
6. Figure 8 highlights that the recovery for both the winning WMH model and the best RLWM model is not optimal for the EEG and Dev datasets. This poor model recovery suggests that the models may not adequately capture the complexities or temporal dynamics inherent in the EEG and Dev data. Additionally, parameter recovery for alphaRL in the Dev and EEG datasets exhibits noticeable noise. While Spearman's rho was used to assess parameter recovery, given the large number of data points (>30), Pearson's correlation might be more appropriate, as it is more sensitive to outliers and better suited for this type of parameter recovery analysis.
7. Figure 12A reveals that the fitted parameters for alphaRL- and alphaRL+ are quite small, indicating that values are integrated gradually over time. This gradual integration suggests that participants may be slowly accumulating evidence, potentially in a manner like reinforcement learning (RL) processes. Does this observation contradict the author's conclusion that no evidence of reinforcement learning was detected in reward-based learning? Addressing this discrepancy could help clarify the interpretation of the results.
8. The possibility mentioned in the discussion that corticostriatal loops typically associated with RL support WM learning is consistent with animal work from Witten and others.
9. As an additional discussion point, adopting the model-free vs. model-based MDP terminology from Hamrick, COBEHA, 2019, which includes state inference in the "model-based" component, really highlights the issue that unless we exactly know what representations are being updated during action selection, it is hard to rule out model-free RL.
10. Readability comments:
 1. Typo on end of page 2, missing word "memory" after "working".
 2. The r_0 parameter could be foreshadowed earlier in the paper.
 3. In Figure 2, it is not clear on first look whether this is all produced by the model or also includes real data; suggest adding a dashed line as part of the legend.
 4. In Figure 3, consider putting all models on the same line in (I am perseverating on the visual code from the preceding figure, in which models are shown on the x-axis of the plot grid ;)
 5. Also in Figure 3, the labels for the model comparison graph are v. confusing for the reader without a knowledge of the r_0 parameter's meaning, consider re-naming?
 6. Why ref Fig. 4 before Fig. 3?
 7. Readability and interpretability of Figure 10 would be improved by labeling panels A, B, and C, and providing a description of the result visualized in each. Additionally, across figures that include visualizations by set-size, the legend should be labeled to remind readers that "ns" refers to set-size.
 8. The implementation hypotheses presented in Figure 5 may extend beyond the scope of the current paper. While these hypotheses are interesting, they make a broader claim about the mapping between algorithmic processes and their neuronal implementation, which the present findings do not empirically support and detract from the focus on behaviorally-driven algorithm discovery.
 9. In the Model Comparison: Exceedance Probability section, it should say '(datasets 1-6).m

Version 1:

Decision Letter:

Our ref: NATHUMBEHAV-24104067A

4th July 2025

Dear Dr. Collins,

Thank you for submitting your revised manuscript "RL or not RL? Parsing the processes that support human reward-based learning." (NATHUMBEHAV-24104067A). It has now been seen by the original referees and their comments are below. As you can see, the reviewers find that the paper has improved in revision. We will therefore be happy in principle to publish it in Nature Human Behaviour, pending minor revisions to comply with our editorial and formatting guidelines.

We are now performing detailed checks on your paper and will send you a checklist detailing our editorial and formatting requirements within two weeks. Please do not upload the final materials and make any revisions until you receive this additional information from us.

Sincerely,

[REDACTED]

[REDACTED]

[REDACTED]

Nature Human Behaviour

Reviewer #1 (Remarks to the Author):

I appreciate the author's efforts in addressing my comments as well as those from the other reviewers. While certain limitations remain, as described below, I consider these to be directions for future research, and the authors themselves have provided sufficient discussion of these issues within the manuscript. Therefore, I am pleased to recommend this paper for publication in its current form.

I agree with the author's claim that "the analysis of the error patterns is completely model independent." Indeed, the fact that the error patterns support the best-fit model and reject competing alternatives is a significant and meaningful contribution. That said, it is still possible that other models could explain the observed error patterns, and thus the current results do not constitute definitive evidence that WM and H systems are directly responsible for guiding behavior. Ideally, a more direct approach—such as regression analysis—would allow the effects of past choices to be explicitly quantified. I understand that such analyses can be difficult to implement, especially in the current task setting with three choice options. The authors acknowledge and discuss these limitations thoroughly, and I consider their treatment of this issue to be appropriate for the present manuscript.

Replacing the exceedance probability with a histogram showing the number of participants best fit by each model using AIC is a simple but nice idea. Like the authors, I also find that BIC tends to favor overly simple models, and that exceedance probabilities can sometimes be overly skewed toward a single model, making it difficult to account for heterogeneity across individuals. In contrast, the AIC-based histogram approach, while relatively simple, may be a useful alternative. At the same time, I believe that methods for model selection and reporting in computational cognitive modeling still vary across researchers, and the lack of standardization is an ongoing challenge in the field. The development of guidelines for model comparison and reporting may be necessary.

Reviewer #2 (Remarks to the Author):

The author has addressed all of my comments well. I have no further comments.

Reviewer #2 (Remarks on code availability):

Code appears to be well organized and commented.

Reviewer #3 (Remarks to the Author):

I appreciate the author's clarification that these findings are specific to a reinterpretation of model-free RL findings,

particularly as they would typically be mapped to the brain.

Thank you also for clarifying the methodological point regarding the WM component being jointly fit (and thus constrained) across set sizes. That is an interesting point that I had not considered on the initial read.

Finally, in conversation with the author, I also am now better appreciating the key point, which is that this paradigm is at least one instance in which model-free RL is not necessary to explain behavior (contrary to what we would have expected from the “consensus” in the literature). Whether this is the case in other task contexts remains a thought-provoking question which this paper leaves us with.

I have no further questions - thank you for this important paper!

I would like to thank all three reviewers for their thorough review of this article. Please find below point by point answers to the reviews, **in purple**. Changes to the manuscript are highlighted in **blue**.

REVIEWER COMMENTS:

Reviewer #1 (Remarks to the Author):

This study represents the culmination of the author's extensive body of research on the role of working memory in reinforcement learning tasks. It provides important evidence that challenges the common assumptions about RL models in reward learning tasks, which have been widely accepted for the past two decades. The methods used in the study are solid and careful, and the results are meaningful. I believe this paper has the potential to make a big impact and is certainly worthy of publication in Nature Human Behaviour.

On the other hand, the current analysis relies on comparisons within a somewhat restricted model space and is dependent on the assumptions of the models. In order to provide more robust support for the author's claims, I would like to suggest several points for further investigation and discussion. In addition, I believe that a more careful discussion of how this study fits into the context of previous research would strengthen its contribution. I outline my suggestions below.

Major point #1

The author proposes a WM model where the learning rate for rewarded outcomes is fixed at 1, and the decay rate for unchosen values is controlled by a parameter, ϕ_{WM} . In contrast, the RL module treats the learning rate as a free parameter and assumes that unchosen values remain unchanged.

Several previous studies have proposed RL models that incorporate forgetting or decaying mechanisms, such as the forgetting-Q learning models, where unchosen values decay at the same rate as the learning rate, or where the decay rate is independent of the learning rate. These models have consistently been shown to fit behavioral data better than models without forgetting or decay (e.g., Ito & Doya, 2009; Gershman, Zhou, & Kommer, 2017; URLs listed below).

<https://pubmed.ncbi.nlm.nih.gov/19657038/>

<https://direct.mit.edu/jocn/article/29/12/2103/28731/Imaginative-Reinforcement-Learning-Computational>

How well would the author's data fit if an RL module incorporating such forgetting mechanisms is used? Would the **WM module with a fixed learning rate of 1** and a choice kernel still outperform such models? Furthermore, if the author's claims are correct, does this imply that the results reported previously with forgetting-Q learning could be reinterpreted as reflecting WM processes? Could the improved fit of models

with forgetting mechanisms be attributed to WM?

In order to clarify the position of this study within the broader context of RL research, I suggest that these points be explored and discussed.

Thank you for the suggestion. We fitted the suggested model (RLwith forgetting + an “H” module = RLfH in figure below) and found the fit significantly worse. This is expected from previous findings (e.g. Collins & Frank 2012) which show that forgetting on its own (without modeling WM-like capacity) is insufficient to capture the size of the set size effect in behavior. A variant of this model that lets the mixture be set size dependent corresponds to the model where WM has a free learning rate parameter (is thus more RL-like), which we had previously fitted and shown also does not improve fit (Fig. 7). We have now added the new RLfH model to the supplementary figure 8, copied below for the reviewer’s convenience.

As the reviewer mentions, we indeed think that it is likely that forgetting-Q models capture some variance from WM processes, accounting for better fits. We show some evidence of this in our recent preprint (Senta et al., 2025, submitted psyRxiv); however, as we find that this is not key to the point made in this paper, as well as somewhat speculative, we have not added this to the manuscript. We are happy to do so should the reviewer think it needed.

Major point #2

The author’s claims basically rely on the goodness-of-fit of the proposed models. While not strictly necessary, supporting these claims with model-neutral methods would strengthen their validity. According to the author, human choices in the task can be explained by two independent effects: a reward-dependent working memory (WM) component and a reward-independent perseveration component based on past choices. It would be ideal to demonstrate the independence of these components and the validity

of each in a manner that does not depend on the specific model assumptions.

We note that the analysis of the error patterns is completely model independent. As such, our results do not uniquely depend on a quantitative “goodness-of-fit” measure comparing models. Instead, following best practices (Palminteri et al 2017), we sought to falsify predictions from distinct theories (WM+RL, vs. WM+H) by showing that the qualitative pattern of results observed in the error analysis matches predictions of WMH, but not of WM+RL.

A simpler alternative might be to use a regression model with predictors representing the choice history at each lag and the reward history for each option (cf. Lau & Glimcher, 2005). This could show that rewards primarily influence recent trials while past choices have effects over longer lags. Additionally, confirming the absence of interactions between reward (given choice) and choice history would further validate the claim (cf. Katahira & Kimura, 2024).

<https://pubmed.ncbi.nlm.nih.gov/16596980/>

<https://link.springer.com/article/10.1007/s42113-022-00145-2>

In addition, if the coefficients for each lag of the choice history can be shown to decay exponentially, it would support the validity of the choice kernel assumption.

However, given the limited number of trials per stimulus in the current dataset, such analyses may lack sufficient data to produce stable estimates. At the very least, discussing these possibilities as directions for future research would be valuable for readers.

The reviewer’s suggestion makes sense. We had indeed attempted a regression-based analysis similar to that suggested by the reviewer. However, such an analysis needs to consider the reward effect as a function of lag in two separable ways (stim-dependent iterations vs. stim-independent trial numbers, which impact working memory), in addition to the choice history effect; furthermore, choice history is strongly correlated with reward history; finally, the presence of 3 choices (instead of two as in the referenced papers) further complexifies the analyses. We found that the more direct error analysis (which does not rely on statistical assumptions like a regression analysis) led to more interpretable results that did provide a model-independent, qualitative test of competing theories.

We have added this as possible direction for future research in the discussion and cited the appropriate papers.

Major point #3:

The estimated value of the WM decay rate, ϕ_{WM} , is not explicitly reported, and the author should provide this information. Related to Major points #1 and #2, if ϕ_{WM} is close to 1, it could be interpreted as a special case of the Forgetting-Q learning model

where the learning rate equals the forgetting rate. In such a case, the effects of past rewards at each lag might manifest only as independent main effects, with no interactions (e.g., no interaction between rewards from one trial back and two trials back).

If ϕ_{WM} takes a value different from α_{WM} , interactions between past histories would be expected to emerge. This distinction is important, as it directly relates to whether the effects of past rewards are additive or interactive (cf. Katahira, 2015).

<https://www.sciencedirect.com/science/article/pii/S0022249615000218>

We would like to thank the reviewer for this note. While fit parameters were reported implicitly in supplement (Figure 9), to improve clarity we have now added a new supplementary figure (Fig. 11) plotting parameter distributions separately. ϕ_{WM} is typically in the [0,0.2] range, thus far from the value of the WM learning rate (1). Due to the size of the figure, we do not incorporate it here; it can be found p45 of the revised manuscript.

Minor points

p.2

“for example, perseveration strategies might be mistaken for a learning rate asymmetry in RL (Sugawara & Katahira, 2021),”

- The original study making this claim is Katahira (2018), and it should be cited in this context.

<https://www.sciencedirect.com/science/article/pii/S0022249617302407>

Apologies for this oversight, we have changed the reference.

p.18

In the Model fitting section, it is stated that `fmincon` was used to optimize the log-likelihood. However, it is not clear whether independent maximum likelihood estimation was conducted for each participant (without any prior). Since protected exceedance probability is calculated, this seems likely, but this assumption might not be obvious to all readers without prior knowledge.

We have now clarified in the methods section that this was performed independently for each participant.

In addition, if independent maximum likelihood estimation was performed, how were the log-marginal likelihoods (log-model evidence), required for computing the protected exceedance probability, calculated? It is possible that BIC was used as an approximation, but this should be explicitly stated for clarity (while some researchers use AIC, I think that AIC was not derived as an approximation of marginal likelihood and is therefore not suitable for this purpose).

Thank you for this important point. We had indeed used AIC, and the reviewer is correct that AIC is not theoretically appropriate as log-model evidence for exceedance probability.

In our experience, BIC strongly overpenalizes more complex models in this type of task environment, as indicated by model recovery analyses (such as figure 8 in the supplement). Indeed, we find that using BIC in this study leads to systematic selection of simplest models that are falsified at validation (i.e. cannot capture the behavioral patterns); we also find that BIC cannot recover ground truth models in simulation studies, while AIC can. As such, using BIC here is not a satisfactory option.

We have now replotted quantitative model comparison results to show mean/error bars of AICs as well as proportion of best fit participants, instead of exceedance probability, and justified this approach in the methods. Modified figures include fig. 2 (copied below as an example), 3 and supplementary model comparison figures. Our conclusions do not change.

p.20

The “s” representing stimulus in the equations is in mixed case.

Thank you, s is now all lower case.

p.25-

The number of significant digits in the tables should be standardized to an appropriate level to improve clarity and consistency. Additionally, the representation of p-values should adhere to standard conventions, such as reporting exact values (e.g., 0.104) or thresholds (e.g., < 0.001).

Good point, we have now adjusted the table to be consistent with number of significant digits and p-values.

Reviewer #2 (Remarks to the Author):

Summary

This paper builds on a substantial body of prior work by Collins, demonstrating the role of working memory in reward-based learning. Previous studies suggested that when individuals solve reinforcement learning (RL) tasks (e.g., learning the correct action for a given stimulus through trial and error), their behavior reflects multiple processes rather than just simple RL algorithms (e.g., updating a stored action value for each stimulus according to a prediction error). In addition to the RL component, which learns slowly, a working memory component is also involved. This component is modeled as an RL agent with a high learning rate, subject to forgetting over time, and its influence is limited by set size. This new paper re-analyzes six datasets previously thought to support this model and re-investigates the contributions of the basic RL component, showing that it is better described as a perseverative associative process (reinforced in the direction of the previous action, regardless of whether that action was rewarded). The paper concludes that there is no simple or typical “RL” component in human reward learning, contrary to expectations based on dopamine and value updating theories.

Overall contribution

I’m somewhat undecided on the overall importance and information gain of the findings. In comparison to the prior body of work by the author, this new article argues that there is no incremental reinforcement learning component in human reward learning. Rather, there is just a working-memory-related RL component and also a perseverative component. This is taken to argue against simple reductionist accounts of RL which perhaps would assume that because dopamine represents reward prediction errors and neuroscience theories cast value learning as gradual synaptic updating, then behavior would also reflect this. I think this argument is quite important.

A. Having said that, I have some hesitations about the strongest arguments that this paper puts forward, and also what the importance is of these particular findings in advancing this argument. First, I don’t think the findings show that RL models of human behavior are incorrect, or that humans don’t do RL in solving these types of tasks. Rather, I think they simply show that human RL involves working memory.

Thank you for your careful read of the paper, and for this point. As noted by us and multiple reviewers, the term RL is often used in different ways, which leads to confusion. We of course agree that human RL – the behavior of humans learning from reward reinforcements – involves working memory. However, we think our results support a more precise conclusion: that, in this experimental context, this behavior is best explained as a mixture of two algorithms that, separately, do not straightforwardly fall into the typical understanding of what an RL algorithm is, because they cannot on their own lead to a policy that optimizes future reward. On its own, the WM process defined by our model only leads to an optimal policy under extremely low loads; on its own, the H cannot lead to an optimal policy (figure 4).

We have added more discussion clarifying what RL is, and we have made sure to refer to the human behavior as “reward-based learning”, to further help clarify. Additionally,

we have attempted providing more nuance on our conclusions about the role of RL mechanisms in human reward-based learning more broadly.

B. Second, the existence of a perseverative or “H” component already is a standard feature of models of human bandit learning. For example, this type of model is used in standard bandit tasks (e.g. Rutledge et al., 2009, Journal of Neuroscience) and also in the two-step task (Daw et al., 2011, Neuron - note here that the perseverative component is not just for successive motor actions). Interestingly, the new model in this paper winds up looking more like the previous model in these papers - having a component that does RL with a somewhat high learning rate and also a perseverative component. In relation to these models, the new model in this paper adds that the RL process has WM characteristics.

I fully agree with the reviewer that the idea of perseveration is not novel, a point we acknowledged in the original submission, and further emphasize in this revision.

However, it is striking that this component is typically considered a “nuisance” factor, and introduced to absorb variance from what is thought of as a low-level process that does not meaningfully contribute to cognition; for example, Rutledge et al 2009 (thank you for pointing to the paper, which we now cite) labels this process as unrelated to the learning process.

By contrast, an important contribution of our paper is to show that this H/perseveration process is integral to learning a good policy, and to show evidence that it operates in parallel with WM-dependent processes to contribute crucially to creating a relevant, reward-optimizing policy. We argue that this finding is novel.

C. I do think the demonstration that the only RL behavior that is detectable in these tasks is influenced by (and may be entirely constituted by) working memory processes is valuable. And I agree with the argument that this argues against a reductionist view that human RL behavior reduces to incremental synaptic-like updating by dopamine. To some extent, though, I think this is already appreciated by the field. For example, Wang et al., 2018, Nature neuroscience (prefrontal cortex as a meta-learning system), demonstrate computational models for how RL computations can take place by activity rather than weight-based changes (notably here, learning the RL strategy itself involves dopamine). More generally, in the field, with the rise of transformers and language models there is now an increasing appreciation that strategies like RL can have a multiplicity of implementations (in weight vs activation based like WM). Nevertheless, the current article is a nice demonstration in this direction. I wonder though whether the position it argues against, and the relative portrayed dominance of this position, is somewhat overstated.

We agree that there is growing awareness that learning can result from non-typical RL processes; nonetheless, how to disentangle them experimentally, what these processes are, and how they work together is still very much understudied, and the weight given to

RL-like processes, as opposed to processes implementing different algorithms, remains extremely high, particularly in human studies. We also note that research on non-typical RL processes overwhelmingly focuses on contexts with significantly more complex decision problems than the simple associative learning ones we consider here.

I'll now move to specific critiques of the analysis.

Analysis critiques

1. It is worth noting that the perseverative component of RL models can sometimes result from mis-specification of the RL model. For instance, Findling/Skvortsov et al. (2019, Nature Neuroscience) demonstrate that noisy RL updates to values can produce behavior that appears perseverative (e.g., behavioral correlations across successive decisions). Can this be ruled out as a reason for perseverative behavior here?

Thank you for this thought-provoking point. Indeed, the Findling et al paper shows that noisy updates in proportion to the RPE lead to choice autocorrelation that is similar to a perseveration bias. We note that the very high learning rates in the Findling paper and the fact that participants learned about a single pair of shapes makes it highly likely that participants used some WM processes to solve the learning task, rather than corresponding to the slow process in our task. Nonetheless, we agree with the reviewer that this possibility should be considered; unfortunately, due to the intractable nature of the likelihood of the noise models, and the limitations of existing techniques used for fitting likelihood intractable models (including our own and the one used in Findling et al), it is not possible to fit such models to our data and to compare it directly to other models. We now note this as a limitation in our discussion.

2. Relatedly, can a three-component model that includes RL in addition to WM and H components be ruled out?

A three-component model cannot be fully ruled out. However, the influence is minimal enough during learning to not counteract the learning to avoid errors. To verify this, we fit such a three component model family (see fig. S8, WMRLH model), and we observed that it fit worse than WMH only. We note that we nevertheless do not rule out this model because it can capture behavior adequately (as expected since the winning WMH model is nested in the three component model). We have clarified the discussion to further emphasize that we think a three-component model is not unlikely, given many other findings in the literature.

3. Could there be general action biases for a stimulus (e.g. action 1 arbitrarily preferred for stimulus 1)??. Could these mask as a perseverative effect?

Strong action biases would indeed mask as perseverative effect, making the corresponding error more likely irrespective of feedback. However, we expect that they would correspondingly lower accuracy. To test the possibility that our error pattern is due to biases, we fit the original family of models (RL+WM with varying learning rate biases) with an additional action bias, where the action-bias policy replaces the uniform random policy in the mixture:

$$\pi_{RLWM} \leftarrow (1-\epsilon)\pi_{RLWM} + \epsilon\pi_{bias}$$

We found that the best fitting model of this family does not fit better than the best fitting model of the original RLWM family in any data set (as measured by AIC; see fig SX). More importantly, we find that this best-fitting model cannot capture the behavior qualitatively, with significant underfit in learning curves and not replication of the key pattern of errors. (see fig. S7, “bias” model).

4. Given that inferences about the slow process rely on factoring out the fast process, it is necessary to interrogate whether the WM process is correctly modeled. I have the following questions and suggestions about the WM part of the model.

First, if participants are aware that there is only one correct action per stimulus, then I would think a WM component would do counterfactual updating (e.g., learning that action 1 is correct would imply action 2 is incorrect). Has this model been tried? Are results around the slow model robust to this?

Participants are indeed instructed that there is only one correct action per stimulus and may perform to counterfactual updating. WM with counterfactual update, however, leads to updates that are not distinguishable from WM without counterfactual update in our context: because the WM updates are designed to lead to one-shot learning, and because the weights of different actions are compared to each other for choice, additionally downweighing unchosen actions after a reward would not materially change the derived WM policy.

5. It appears that capacity parameter K is constrained to be an integer. This seems unjustified. What happens if capacity parameters are allowed to vary continuously? More generally, I think K should also be unbounded. Based on the current setting, I think K being bounded at 5 prevents the WM system from ever fully turning off.

We indeed constrain capacity K to be an integer. Our arguments to do so are:

- This is in line with the “slot theory of WM”, which is a logical approach in the context of discrete chunks of information to be stored here (s-a associations)
- It helps make capacity interpretable.
- Pragmatically, because the set sizes are discrete, there is no identifiability for a more continuous version of capacity, and the non-smoothness of the likelihood function at discrete points caused issues to optimizers in the absence of priors.
- In situations where we have used priors and thus were able to compare discrete vs. continuous encoding of WM capacity, we have always found the same conclusions. (e.g. Rmus et al, 2023, Frogner et al, 2025, Senta et al, submitted, other unpublished).

As such, we clarify that discrete capacity is not a theoretical commitment of our approach, but a pragmatic implementation which we have previously confirmed does not impact findings. We have made a note of this in the modeling methods.

The bounds on K also reflect a mixture of theoretical prior and pragmatic constraints. Most literature agrees that most participants can hold at least 2 independent chunks in memory, and that few can hold more than 5 (e.g. Oberauer et al) – this is indeed what we consistently observe in our study.

Pragmatically, given the distribution of set sizes 2-6 (chosen because of the known typical distribution of WM capacities), a capacity value above 5 predicts next to no set size effect, and in practice makes the modules difficult to identify, leading to model degeneracy. Any capacity value above 6 will have the same exact likelihood, so bounding it is necessary.

We note that this constraint does not prevent from turning WM off. WM can be turned off with the “rho” parameter set to 0; a higher capacity would simply equate the amount of WM contribution to the policy across all set sizes.

We are open to re-running all fits with either continuous capacity or with inclusion of a capacity parameter of 1; however, given the high computational cost of doing so, and our strong prediction that it cannot improve fit (informed both by theory and prior experience), we would prefer not doing so, unless the reviewer specifically requires it.

6. Related to the above, it is somewhat hard to understand why an RL agent with no learning from negative feedback cannot capture the basic error analysis effect. The paper reports that this is because the WM model remains active even at very high stimuli. This raises the question of why the WM component cannot turn off at a high number of stimuli. Could this be due to the parameter constraints mentioned above?

Indeed, without WM, an RL model with only gain learning rate would predict the high set size error pattern. Our model does assume that WM doesn't fully turn off at a high number of stimuli – this is consistent with most WM literature which shows that participants still use WM when above capacity, but simply store a subset of information. In our case, with a capacity of 2-4, participants might still be able 33-66% of information (relative to set size 2) in set size 6. Nonetheless, we agree that this assumption is untested in this specific modeling. Thus, we additionally fit the original model family (RLWM), but modified the WM contribution to assume no WM role when above capacity (i.e. if $n_s > WM$, the policy is fully determined by RL). We find that the best model under this assumption fits significantly worse than the best RLWM model, and cannot capture the pattern of behavior in learning curves or error patterns well. See supplementary figure S6, model “WMdrop”.

7. In the mixture model, averaging occurs at the level of the policy. In other similar mixture models (e.g., model-based / model-free by Daw et al., 2011), it is the value predictions of the two components that are averaged. What is the justification for averaging the policies rather than the values? Is this supported by model comparison?

In such frameworks as MB-MF (Daw et al 2011), both processes are proposed to estimate the same quantity (expected value) using different methods, so mixing at the value level makes sense a priori. Here, the processes have no reason to be thought of as being on the same scale/in the same “unit” (e.g. the RL module estimates a value, but the WM module computes association weights). As such there is no clear reason why they should be mixed directly. Our approach treats them as separate processes

that could independently generate a choice, and a mixture that serves as an arbitrator. We note that there is also a large number of precedents for such “mixture of experts” types of approaches in the literature. We have added this as a note in methods.

Minor

8. Is the printed equation for the mixture weight a mistake ($p_{wm} = p \cdot \min(1, \frac{ns}{K})$)? I would think capacity should be divided by the set size (divided evenly among the items) rather than the other way around.

Indeed, thank you for catching this mistake.

9. Please list the best-fitting parameters of the key models. It would be good to verify the fitted learning rates of each system.

We have now added this to the supplementary information (Fig. 11). The learning rates are very consistent across models and data sets, with low values.

10. For each model, please clearly list all parameter constraints in the fitting process and provide some consideration on how it is considered that such constraints do not affect interpretations.

The constraints are now included in the Computational modeling methods (model fitting section). Because for most parameters, we do not have extra constraints, we do not comment on those. We have added information regarding the working memory parameter.

Reviewer #3 (Remarks to the Author):

In this paper, Collins presents model-based and empirical evidence against a widely held assumption in instrumental tasks that model-free RL (defined as updating action values using a delta rule in response to RPEs) contributes to choice behavior. The author employs an error-based analysis across seven datasets, all from a paradigm designed to dissociate the contribution of working memory and model-free RL. The author also develops a novel alternative model to explain the error patterns. The key finding is that it is sufficient to include an outcome-independent habit mechanism, in addition to outcome-sensitive WM, to explain the error patterns. I broadly agree with the author's point that even "simple" learning tasks mask the presence of other cognitive processes that we cannot measure, but which nevertheless contribute to choice behavior. The novel demonstration that when those are factored out, WM + habit is enough to explain behavior in this task is compelling, and will be of wide interest to the community studying related tasks. That said, I am not convinced that the findings generalize as broadly as the paper claims. Below are some comments and questions:

1. The term "RL" is overloaded in the paper/title. It is introduced, as is common, as a framework that can explain a variety of learning processes. But in this paper (and

indeed maybe the field at large), RL is equated with a much narrower definition of model-free RL (e.g. on page 3) that updates values using a delta rule. Not being consistent & precise in terms of terminology might lead readers (especially those new to the field) to get the impression that RL as an explanatory framework is narrower than it actually is. In reality, there is broad consensus that, as the author clearly shows in the RLWM task, model-free RL on its own is insufficient to explain behavior.

Indeed, the word RL is overloaded in the broad literature, as we agree here and elsewhere (e.g. Eckstein et al Collins RL OECS, etc.). For that reason, the article uses “reward-based learning” consistently to refer to the human behavior, and carefully specified the type of “model-free RL processes” we considered, which is the one typically considered to map best to underlying neural RL mechanisms; and also because it is the type of process used to model contextual bandits tasks, as considered here. We have clarified this in the introduction and expanded the discussion to further bring attention to the importance of consistent & precise terminology around RL.

2. The second suggestion is to consider what happens when we recast WM and other functions that work in service of model-based learning (e.g. attention), as internal actions (Dayan, 2012; Callaway and others in recent years). In this sense, actions are clearly not outcome independent, as success depends on using the correct model (i.e. loading efficient representations in WM, directing attention to relevant features, hypothesis sampling etc). Such actions might need to be learned (see work on attention learning from Kruschke, Niv and others). Can we confidently say, based on these data, that "mental actions" are not learned in a model-free way (Bramlage & Cortese)? Or that, once the correct model is considered, learning does not proceed in a model-free manner (Lee ... Witten, Daw)?

Indeed, the reviewer is correct that internal actions, such as working-memory gating, may be learned through model-free RL, and thus that model-free RL processes may be relevant to other processes that support fast and flexible learning, in a different time-scale. This point is made in the discussion. However, this point does not minimize the point we make here – that learning in this experimental context can be explained without direct contributions of model-free RL-like algorithms.

3. This representation learning reframe suggests that an alternative explanation for outcome insensitivity at large set sizes might be "meta-learning", e.g. people verbalizing and updating a concise model/construal/representation of the task (see work from Ho, Radulescu & others). This type of “efficient” model learning is exactly the kind that might be diverging in SCZ and development.

While we broadly agree with the reviewer that it is possible that a form of meta-learning is happening, it is not obvious to us how outcome insensitivity might result from it and how this might be specific to higher set sizes, without a more quantitative characterization of what this task representation might take the shape of.

4. With regards to generalizing to probabilistic versions of task, maybe there is a bit of

an overgeneralization here? The probabilistic version of RLWMP has fixed rewards probabilities, which means participants can still solve it using a policy that might take the form "when this symbol appears I should click this". OTOH Bornstein shows effects of episodic memory on choice in a drifting 4-arm bandit, which seems more convincing. In any case, in real-world learning scenarios, outcomes or optimal state-action mappings are rarely deterministic or fixed; instead, they often involve a mix of uncertainty and variability that challenges purely habit-based or one-shot responses and requires adaptive learning strategies. To strengthen the findings, it would be valuable to test a broader range of task conditions with varying set sizes and outcome stochasticity to understand the interaction between set size and state uncertainty.

Thank you for your comment. In response to the reviewer's point that we overgeneralize the findings with respect to the probabilistic version of the task, we have edited the paper to ensure that the findings are interpreted as precisely within the task context as we can, and not overstated; in particular, we have edited the title of the corresponding results section to be more specific.

While we agree that further generalizability to a broader class of probabilistic settings is important in the long term, we strongly believe it is beyond the scope of this paper: indeed, we note that doing so would require an experimental and computational way of disentangling WM contributions in many different contexts, something that is a highly complex project of its own.

5. Within the context of the data in this paper, one idea would be to compare models across each set size separately. For instance, smaller set sizes (1 to ~3) might involve more one-shot learning, which might make the WMH model fit better due to task demands, rather than an absence of RL-like processes. This would allow a more precise view of strategy use across different task conditions.

The reviewer is correct that learning in small set sizes involves more one-shot learning, as evidenced by the near optimal learning curves (i.e. reaching asymptote within 3 iterations). However, for that reason, learning in the context of low set sizes would be best explained by a much simpler model, since very little learning variance needs to be explained, and as such, they do not bias towards either WMH or WMRL (but rather towards WM only).

Fitting set sizes jointly is what enables us to factor out the WM component from slower learning components (whether RL or H) in our model fitting. The joint model identifies the fast learning component constrained in low set sizes, and adjusts the slow learning component to explain in high set sizes what cannot be explained solely by WM. As such, while a greater investigation of small set size behavior is certainly interesting it is not well suited to the methodological approach applied here and therefore would not help highlight the nature of the slower learning component, which is more present in higher set sizes.

6. Figure 8 highlights that the recovery for both the winning WMH model and the best

RLWM model is not optimal for the EEG and Dev datasets. This poor model recovery suggests that the models may not adequately capture the complexities or temporal dynamics inherent in the EEG and Dev data. Additionally, parameter recovery for alphaRL in the Dev and EEG datasets exhibits noticeable noise. While Spearman's rho was used to assess parameter recovery, given the large number of data points (>30), Pearson's correlation might be more appropriate, as it is more sensitive to outliers and better suited for this type of parameter recovery analysis.

The reviewer is correct that model parameter recovery is not perfect in all data sets, including EEG and Dev. However, we believe this does not undermine our findings for the following reasons.

- We understand why the recovery is weaker in the specifically mentioned data sets. In dev, the experiment is shorter (10 blocks) and the maximum set size is 5 (instead of 6), to make the task appropriate for 8-yo children. The lower amount of data and lower load range decreases the model identifiability as expected. In the EEG data set, participants' performance was particularly high (possibly due to experimenter presence and more commitment than in a purely behavioral data collection), decreasing the variance to be explained compared to when learning is slower.
- We note that model validation is equally good in these data sets as in others, showing that the model does capture performance well.
- None of our arguments are dependent on perfect parameter recovery: the key findings of the paper instead rely on the model's ability to capture qualitative patterns of behavior compared to competing models, not on the value of parameters.
- Indeed, these qualitative findings are similar across all 6 data sets despite experimental protocol variance, making the findings stronger. By contrast, the competing models cannot explain the patterns.

Regarding our use of Spearman correlation, most of our parameters are bounded (or discrete) and distributed in a highly non-normal way, including parameter values at the bounds, violating assumptions for Pearson's correlations. Nonetheless, we checked that findings are similar with parametric tests. Given that parameter recovery is not important to the core findings of the paper, we did not add this to the paper, but are willing to do so should the reviewer find this essential.

7. Figure 12A reveals that the fitted parameters for alphaRL- and alphaRL+ are quite small, indicating that values are integrated gradually over time. This gradual integration suggests that participants may be slowly accumulating evidence, potentially in a manner like reinforcement learning (RL) processes. Does this observation contradict the author's conclusion that no evidence of reinforcement learning was detected in reward-based learning? Addressing this discrepancy could help clarify the interpretation of the results.

Indeed, the low learning rate parameters indicate a process slowly integrating evidence. However, the evidence integrated by the H process is not outcome (as typical of a

model-free RL process), but choice (in a Hebbian or habit-like way). This is evident in the simulation study (figure 4), which shows that in the absence of a secondary, more standard learning process, the H process cannot learn a reliable policy (light blue curve, $\rho_{WM}=0$).

8. The possibility mentioned in the discussion that corticostriatal loops typically associated with RL support WM learning is consistent with animal work from Witten and others.

Thank you for pointing this out, we have added some more animal work citations on this topic in the conclusion, including Engelhard et al 2019.

9. As an additional discussion point, adopting the model-free vs. model-based MDP terminology from Hamrick, COBEHA, 2019, which includes state inference in the “model-based” component, really highlights the issue that unless we exactly know what representations are being updated during action selection, it is hard to rule out model-free RL.

We agree with the broad point that state inference can be seen as a model-based component over which model-free value estimation can occur, leading to more complex behaviors than typically considered model-free. Indeed, this is a point we ourselves have made in a few opinion or review publications (e.g. Cockburn & Collins 2019, Yoo et al, etc.). However, it is not clear to us in this specific case how state inference might rescue the non-RL interpretation of our findings. Nonetheless, we interpret the reviewer’s comment as an additional argument to further soften the possibility of our findings being interpreted as “no-RL”. We have further expanded on this point in the discussion.

10. Readability comments:

Thank you so much for the detailed read and for making notes of those points, we have fixed the points below.

1. Typo on end of page 2, missing word "memory" after "working".

Fixed, thank you.

2. The r_0 parameter could be foreshadowed earlier in the paper.

We were not sure where this would be appropriate. We hope the clarification of other parts of the paper make this now more understandable.

3. In Figure 2, it is not clear on first look whether this is all produced by the model or also includes real data; suggest adding a dashed line as part of the legend.

This is a great point, apologies for not catching this before submission. We have now clarified this.

4. In Figure 3, consider putting all models on the same line in (I am perseverating on the visual code from the preceding figure, in which models are shown on the x-axis of the plot grid ;))

Done.

5. Also in Figure 3, the labels for the model comparison graph are v. confusing for the reader without a knowledge of the r_0 parameter’s meaning, consider re-naming?

Thanks for this comment. We struggled to find the right place in the trade-off between exactness and interpretability: while more interpretable names help understanding, we worry that they will not be exact and could bias the readers' interpretation. As a compromise, we kept the names (noting that the legend explains an interpretation of parameter r_0), but used a color-coding to help with the interpretation of the parameters.

6. Why ref Fig. 4 before Fig. 3?

Thanks for pointing out, we now refer to Fig. 3 before Fig. 4.

7. Readability and interpretability of Figure 10 would be improved by labeling panels A, B, and C, and providing a description of the result visualized in each. Additionally, across figures that include visualizations by set-size, the legend should be labeled to remind readers that "ns" refers to set-size.

Thank you for the great suggestions. Note that this is now Figure 11, due to the introduction of another supplementary figure. We have now made the suggested improvements to the figures, including labeling, set size, and interpretation.

8. The implementation hypotheses presented in Figure 5 may extend beyond the scope of the current paper. While these hypotheses are interesting, they make a broader claim about the mapping between algorithmic processes and their neuronal implementation, which the present findings do not empirically support and detract from the focus on behaviorally-driven algorithm discovery.

This was introduced at the request of other reviewers. However, we agree that this is speculative and not supported by empirical results, in particular for the neuronal implementation. We have moved it to supplementary discussion.

9. In the Model Comparison: Exceedance Probability section, it should say '(datasets 1-6)'.m

We have now taken out the exceedance probability section, as requested by another reviewer.